# A theoretical case-study of Scalable Oversight in Hierarchical Reinforcement Learning

**Tom Yan, Zachary Lipton**
Machine Learning Department
Carnegie Mellon University
Pittsburgh, PA 15213
{tyyan, zlipton}@cmu.edu

## Abstract

A key source of complexity in next-generation AI models is the size of model outputs, making it time-consuming to parse and provide reliable feedback on. To ensure such models are aligned, we will need to bolster our understanding of scalable oversight and how to scale up human feedback. To this end, we study the challenges of scalable oversight in the context of goal-conditioned hierarchical reinforcement learning. Hierarchical structure is a promising entrypoint into studying how to scale up human feedback, which in this work we assume can only be provided for model outputs below a threshold size. In the cardinal feedback setting, we develop an apt sub-MDP reward and algorithm that allows us to acquire and scale up low-level feedback for learning with sublinear regret. In the ordinal feedback setting, we show the necessity of both high- and low-level feedback, and develop a hierarchical experimental design algorithm that efficiently acquires both types of feedback for learning. Altogether, our work aims to consolidate the foundations of scalable oversight, formalizing and studying the various challenges thereof.

## 1 Introduction

Next-generation AI models are poised to produce sophisticated outputs such as long-form texts and videos, and execute complex tasks as agents. To build these AIs responsibly, we need to better our understanding of scalable oversight: the ability to provide *scalable* human feedback to these complex models [2, 8, 15, 5]. An immediate, key challenge to overcome is the size of model outputs, making it time-consuming for humans to parse and provide reliable feedback on, even with AI-assistance [24, 27, 23]. To this end, in this work, we consider human labelers with bounded processing ability such that accurate feedback can only be provided for outputs below some threshold size. We are interested in answering the question:

> How can we scale this limited feedback to supervise a model with outputs *larger* than this limit?

Verily, this task is difficult without further assumptions. If the model output can only be assessed in its entirety, it is impossible for humans to provide reliable feedback. Thus, we investigate a natural setup that gives us hope to overcome the limitation in feedback — when model outputs have *hierarchical* structure. Hierarchical structure exists in many high-dimensional outputs of interest, including long-form texts (books made up of chapters), videos (movies made up of scenes) and code (main functions made up of helper functions). Indeed, it reflects the way we humans produce many of our most complex creations.

38th Conference on Neural Information Processing Systems (NeurIPS 2024).

To formalize the setting, we study scalable oversight in a goal-conditioned hierarchical reinforcement learning (HRL) setup. Goal-oriented RL is a popular approach that has seen sizable success in leveraging state space structure to overcome sparse rewards over long horizons [16, 17, 10]. Our aim in this paper differs in using this as an entry-point into understanding how to scale up bounded human feedback, and formalizing the conceptual/technical challenges thereof. It turns out that one known advantage of HRL, besides more efficient exploration and efficient credit assignment, is the ability to enable scalable oversight.

## 1.1 Preliminaries

We consider a finite-horizon, Markov Decision Process (MDP) $\mathcal{M} = \langle S, A, P, r, s_1, H \rangle$, with finite state space $S$, finite action space $A$, transition probability $P : S \times A \to \Delta(S)$, reward $r(s, a) : S \times A \to [0, 1]$ and finite horizon $H$. The learner interacts with $\mathcal{M}$ starting at state $s_1$ and the episode ends after $H = H_h H_l$ time-steps. In this work, policies are trained using human feedback. And so, we assume that a human supervisor is needed to evaluate and provide reward $r$ for trajectories $\tau \sim \pi, P$ generated by policy $\pi : S \to A$.

**Accompanying Example:** Consider the task of learning to generate a long-form, argumentation essay. Providing feedback to an end-to-end policy is difficult as labelers would have to read through entire essays to rate the outputs, after which it may be difficult still to assign a single rating to the entire essay. A tractable alternative is to learn a hierarchical model, with a higher-level policy that generates the essay arguments (goals), and lower-level policies that flesh out these points (realize these goals). It would then be easier for the labeler to rate the shorter-length essay content, and also individual fleshed out arguments, in order to generate a rating on the whole. This approach also mirrors existing rubrics for scoring essays [1].

**Bounded Feedback:** To formalize the difficulty of human supervisors assessing long-form outputs, we assume that reliable feedback can only be provided for trajectories of length at most $\max(H_h, H_l)$. In particular, this means that for the global policy $\pi : S \to A$, it is infeasible to obtain reliable feedback for its trajectory $\tau \sim \pi, P$, as $|\tau| = H_h H_l$. This thus motivates hierarchical learning, which makes possible the acquisition of reliable feedback in spite of bounded human supervision.

### 1.1.1 Goal-conditioned HRL

Since we are unable to learn a single, monolithic policy, our goal instead will be to learn a set of smaller policies that make up a hierarchical policy. This set consists of a high-level policy $\pi^h : S \to \Delta(A_h)$ (outputs a high-level action $a^h$ at state $s \in S$), and a set of low-/sub-policies $\pi^l_{s,a^h} : S^l_{s,a^h} \to \Delta(A)$, where $S^l_{s,a^h} \subseteq S$ is the set of all states reachable from $s$ after $H_l$ steps.

In a nutshell, the high-level policy designates goals by choosing high level actions. The low-level policies then aim to realize these goals, while also trying to achieve a high intermediate return. Importantly, both such policies act over a shorter horizon of at most $\max(H_h, H_l)$, making it amenable for human supervisors to evaluate.

**Goal Function:** in the goal-conditioned HRL setting, we assume access to a function $g$ mapping high-level action $a^h$ at state $s$ to a goal-state $g(s, a^h) \in S^l_{s,a^h}$. For example, $s$ is the current content of the essay, $a^h$ is the action (in natural language) "add an argument using X" and $g(s, a^h)$ is the content of the essay with the "argument using X" included.

**Goal-conditioned sub-MDP:** Given a high level action $a^h$ at state $s$, this defines the sub-MDP $M(s, a^h)$, which has state space $S^l_{s,a^h} \subseteq S$, action space $A$ (action space of the original $\mathcal{M}$), transition probabilities $P$ restricted to $S^l_{s,a^h}$, starting state $s$ and finite horizon $H^l$. The sub-MDP reward $r^l$ will be defined later and as we will see, an apt choice is important for achieving sublinear regret.

**High-level MDP:** Given a set of low-level policies, $\pi^h$ may be thought of as operating over a high-level MDP with state space $S$, action space $A^h$, starting state $s_1$ and finite horizon $H^h$. Importantly, the high-level transition $P'$ of this MDP is a function of the current set of low-level policies $\Pr'(s'|s, a^h) = \Pr(s^{\pi_{s,a^h}}_{H_l} = s')$, which denotes the distribution over the (final) $H_l$th state that $\pi^l_{s,a^h}$ reaches. Furthermore, the high-level reward $r^h(s, a^h) = \mathbb{E}_{s_j, a_j \sim \pi_{s,a^h}, P}[\sum_{j=1}^{H_l} r(s_j, a_j)|s_1 = s]$

corresponds to the intermediate return of sub-policy $\pi_{s,a^h}$ in $M(s, a^h)$. Altogether, this gives rise to a key complication in hierarchical learning. This is that both the transitions and rewards in the high-level MDP are non-stationary, as sub-policies $\pi_{s,a^h}$ are updated over time.

**Interaction Protocol:** At each time-step $t$, the high level policy chooses a high level action $a_t$ based on current state $s_t$. This defines the sub-goal state $g(s_t, a_t)$, along with the corresponding sub-MDP $M(s_t, a_t)$ with finite-horizon $H_l$, in which sub-policy $\pi^l_{s_t, a_t}$ is used to try to achieve the goal. The overall return of the high level policy $\pi^h$ and low-level policies $\{\pi^l_{s,a}\}_{s,a \in S \times A^h}$ is the sum of intermediate returns $r(\pi^l_{s_t, a_t})$ incurred:

$$
V^{\pi^h, \pi^l}(s_1) = \mathbb{E}_{a_t \sim \pi^h(s_t), s_{t+1} \sim \Pr(s^{\pi^l_{s_t, a_t}}_{H_l})} [\sum_{t=1}^{H_h} r(\pi^l_{s_t, a_t}) | s_{t=1} = s_1].
$$

**Instantiation in the example:** returning to our example, for a cogent essay, the arguments need to be logically related and built on top of each other. This results in a sequential decision making problem corresponding to the one solved by the high level policy $\pi^h$. Given an argument $g(s, a^h)$ to flesh out, the low level policy $\pi^l_{s,a^h}$ generates up to $H_l$ words, whose content aims to realize this argument. Additionally, low-level policies can incur intermediate rewards (return) for eloquent diction and clear structure when fleshing out the argument, all of which add to the essay's persuasiveness.

### 1.1.2 Learning Task

Our aim is to learn a hierarchical policy, whose return is close to that of the optimal, goal-reaching hierarchical policy, which we define as follows. For brevity, from this point on, we will use $a^h$ and $a$ interchangeably to denote high level action.

**Assumption 1** (Goal-Reachability). *In every sub-MDP $M(s, a)$, there exists a policy that achieves the goal $g(s, a)$ almost surely. That is, there exists at least one policy $\pi \in \Pi_{s,a}$ in the policy class $\Pi_{s,a}$ such that $\Pr(s^\pi_{H_l} = g(s, a)) = 1$.*

In other words, we assume that the goal function $g$ is well-defined in that it designates goals that are feasible to reach from the starting state $s$ (e.g. the argument can be successfully fleshed out in $H_l$ words or less given the essay content thus far). To motivate this assumption, we note there that there are already many settings of interest, where we have prior knowledge of a good goal function. This is because we humans have often (and successfully) taken the hierarchical approach to build up to and produce these long-form creations. So we know what are good goals to set e.g. we write essays by first writing an outline of arguments, then expanding out each point in the outline. Indeed, this approach of explicitly encoding prior knowledge in the hierarchical learning algorithm has been done in both HRL literature (e.g. we know apriori mazes has hierarchical structure in that it consists of rooms [22]) and scalable oversight literature (e.g. we know that books consists of chapters [27]).

With this assumption, there exists constant $C$ large enough such that if $\pi \in \arg\max_{\pi \in \Pi_{s,a}} r(\pi) + C \cdot \Pr(s^\pi_{H_l} = g(s, a))$, then $\pi$ is goal-reaching and $\Pr(s^\pi_{H_l} = g(s, a)) = 1$.

**Definition 1.** *Define optimal low-level policies as $\pi^*_{s,a} \in \arg\max_{\pi \in \Pi_{s,a}} r(\pi) + C \cdot \Pr(s^\pi_{H_l} = g(s, a))$. Define optimal high-level policy as $\pi^* = \arg\max_{\pi \in \Pi^h} V^{\pi, \pi^*_{s,a}}(s_1)$.*

In words, $\pi^*_{s,a}$ has the highest intermediate return of all goal-reaching policies. Now let $\pi^*$ be the optimal high-level policy fixing each sub-MDP policy to be $\pi^*_{s,a}$.

**Learning Goal:** We wish to learn a set of near-optimal high- and low-level polices $(\pi, \{\pi_{s,a}\})$ such that: $V^{\pi^*, \pi^*_{s,a}}(s_1) - V^{\pi, \pi_{s,a}} \leq \epsilon$.

### 1.2 Takeaways

The broad takeaway from this paper is that hierarchical structure, if it exists, can be provably used to scale up limited human supervision. That is:

Hierarchical learning can enable scalable oversight.

On a more technical level, this paper studies the challenge of training a set of (instead of a single) policies that work together to form the hierarchical policy. This is the more complicated problem we turn to solve when it is not feasible to train a monolithic policy, due to bounded human supervision. We thus consider learning in the goal-conditioned HRL setup, under both cardinal and ordinal feedback. A key insight that applies in both settings is that an apt sub-MDP reward design (a suitable penalty for non-goal reachability) is needed for bounding regret and controlling the exit state of learned low-level policies. This is so that learned sub-policies do not land at bad states with sizable probability. Doing so would then allow one to compose low-level policies together, and stabilize high-level policy learning in the high-level MDP. More specific takeaways for both types of feedback are as follows:

- Under cardinal feedback, we develop a novel no-regret learning, Algorithm 1, that jointly learns a high-level and a set of low-level policies. Notably, Algorithm 1 only requires low-level feedback. Our main structural result in this setting is that hierarhical RL reduces to multi-task, sub-MDP regret minimization. Thus, the regret from the low-level accumulates additively (instead of say multiplicatively) as speculated upon in [15].

- Under ordinal feedback, we develop a novel hierarchical experiment-design Algorithm 2, building off of existing work on experiment design in preferenced-based RL [29]. A key observation is that in the ordinal case, low-level feedback may not be sufficient and high-level feedback may be needed. This introduces complications in human supervision, as the high-level feedback would need to account for the *current* performance of sub-policies. To this end, we study two natural forms of feedback, requiring differing cognitive loads on the human supervisor. Through the experiment design algorithm we develop, we then analyze the differing sample complexity under the two types of feedback. Finally, we show that high-level feedback should not be used if low-level feedback is sufficient and one form of feedback, with higher cognitive load, leads to better sample complexity.

## 2   Related Works

**HRL under cardinal rewards:** There has been sizable interest in understanding of the sample complexity of HRL algorithms, which to our knowledge has thus focused on learning from cardinal rewards. On this subject, the two closest papers to that of ours are [22] and [25]. [22] studies goal-conditioned HRL with the key result being a sample complexity lower bound associated with a given hierarchical decomposition. On the upper-bound side, an algorithm (SHQL) is presented, albeit without theoretical guarantees. By contrast, our work presents a learning algorithm with provable guarantees, and further shows that learning in goal-conditioned HRL reduces to multi-task, sub-MDP regret minimization.

[25] studies HRL under the options framework, providing a model-based, Bayesian algorithm with access to a prior distribution over MDPs that is updated over time. It does not adaptively learn sub-policies based on observed returns, computing instead an option for every exit-profile and equivalence class at each time during model-based planning. By contrast, our work does not assume knowledge of the prior nor ability to update posteriors, and does adaptively explore sub-MDPs via the UCB principle. Additionally, [25] demonstrate that when the size of the set of exit ("bottleneck") states is small, learning is efficient. Our work shed further light on this insight by showing that under a suitable sub-MDP reward, we can induce a small set of exit states *with high probability*. Thus, even though the total number of possible exit-states may be high, this condition is sufficient for learning with sublinear-regret.

**RL under ordinal rewards:** There has also been considerable interest in bandits/RL from preferences [26, 28, 18, 14, 30, 29]. Following the demonstrated success of RLHF [9, 31, 19, 4], there has been great interest in studying offline RL from preference feedback, and particularly experiment design for enhanced sample efficiency [30, 29]. Due to the success of RLHF in alignment, we also consider studying scalable oversight in this setup. Please see the Appendix A for further discussions on scalable oversight and goal-conditioned RL.

# 3 Learning from Cardinal Feedback

We begin by considering the setting when feedback is in the form of cardinal rewards. As noted before, in HRL, the high-level policy performance is dependent on the low-level policies performance. Thus, a naive approach is to learn near-optimal sub-policies in every sub-MDP $M(s, a)$, and then learn a high-level policy on top. However, a more sample efficient approach is to strategically explore sub-MDPs, and discover sub-policies with high intermediate returns in tandem with a high level policy that visits these "good" sub-MDPs. Please see the Appendix C for all the proofs. Note that in what follows, for brevity, theoretical statements will contain the phrase "with high probability" and the appendix will contain proofs that formalize this guarantee.

## 3.1 Sub-MDP reward design for Hier-UCB-VI

We are interested in adaptively learning the necessary sub-policies (the useful goals to achieve) and the associated high level policy that invokes these sub-policies. It is natural then to adopt an upper confidence bound approach and construct an exploration bonus that tracks the best/unexplored sub-MDPs. To this end, we develop an adaptation of the classic UCB-VI algorithm [3]. We highlight two key ingredients needed to construct the Hier-UCB-VI Algorithm 1.

**Tradeoffs in sub-MDP reward design:** Learned sub-policies in HRL have to tradeoff between two objectives. One is high intermediate returns $r(\pi_{s,a})$. The other is that exit-state; sub-policies should not land at "bad" states, as even if the intermediate return is high, $V(s_{H_l}^{\pi_{s,a}}) \approx 0$ means the return from hereon out (and hence the overall return) will be low. Thus, in sub-policy learning, we also need to consider the goodness of the exit-state. But how can we incentivize sub-policies to land at "good" states without being able to calculate $V$? Luckily, in the goal-conditioned setting, there is a natural answer for a "good" exit-state: $g(s, a)$.

To operationalize this, we design a sub-MDP reward that trades-off between intermediate sub-MDP return and goal-reachability. In sub-MDP $M(s, a)$, at time-step $h$, sub-MDP reward $r_{l,h}(s', a') = r(s', a') + \kappa \mathbb{1}(h = H_l \wedge s' = g(s, a))$. Crucially, here we set the weighting $\kappa = \max(2H_h H_l, C)$, which corresponds to an upper bound on the regret should we not reach the goal-state.

**UCB construction:** Next, we wish to obtain an UCB for $r(\pi_{s,a}^*)$. Our main observation is that by using a no-regret subroutine for learning in $M(s, a)$, the regret guarantee directly translates to a UCB. Due to our choice of sub-MDP reward $r_l$, the UCB includes a penalty on non-goal reachability.

**Lemma 1** (UCB implied by sub-MDP regret). *Let $UB(\mathcal{R}^n(s, a))$ be an upper bound on sub-MDP $M(s, a)$'s cumulative regret after $n$ rounds. Define $\beta = (\kappa + H_l)2\log(\frac{|\mathcal{C}(S,A)|H_h K}{\delta})$ and bonus,*

$$b_r^{s,a}(n) = \frac{UB(\mathcal{R}^n(s, a)) + \beta\sqrt{n}}{n} - \frac{\kappa}{n}\sum_{i=1}^{n}\mathbb{1}(s_{H_l}^{\pi_{s,a}^i} \neq g(s^h, a^h)).$$

*Then, the average reward plus bonus $\bar{r}_n(s, a) + b_r^{s,a}(n)$ is an UCB for $r(\pi_{s,a}^*)$ with high probability.*

**High-level MDP transition stabilization:** An additional benefit of incentivizing goal-reachability is that we know the idealized transition probability in the high-level MDP. As mentioned before, another key difficulty with HRL is that the empirically estimated transitions in the high-level MDP drifts over time. In our algorithm, the key stabilization approach is to avoid estimation and set the transition in the upper bound $Q_i$ to be the idealized transition ($g(s, a)$ w.p. 1). This allows us to prove our regret guarantee as described below.

## 3.2 Regret Analysis of Hier-UCB-VI

We start with a definition on clusters of equivalent sub-MDPs. Let there be $\mathcal{C}(S, A^h)$ such clusters. In the most general setting, it is not known apriori if there is any shared structure, in which case each sub-MDP will simply be its own cluster.

**Definition 2** (Equivalent sub-MDPs [25]). *Two subMDPs $M(s, a)$ and $M(s', a')$ are equivalent if there is a bijection $\mathcal{F}$ between state space, and through $\mathcal{F}$, the subMDPs have the same transition probabilities and rewards.*

Our main structural result is that HRL regret decomposes to multi-task, sub-MDP regret in the cardinal reward setting. This has the implication that only low-level feedback is needed for regret

---

**Algorithm 1** Hierarchical-UCB-VI (Hier-UCB-VI)

---

1: Initialize $D = \emptyset$, $Q_{H_h+1}(s,a) = H_h H_l \ \forall s,a$, $V_{H_h+1} = 0$, $\kappa = \max(C, 2H_h H_l)$
2: **for** episode $k = 1, ..., K$ **do**
3:     **for** timestep $i = H_h, ..., 1$ **do**
4:         **for** $(s,a) \in S \times A^h$ **do**
5:             **if** $(s,a) \in D$ **then**
6:                 Update UCB: $UB(r^{\pi^*}(s,a)) = \bar{r}_{N^{k,h}(s,a)}(s,a) + b_r^{s,a}(N^{k,h}(s,a))$
7:                 Set:

$$Q_i(s,a) = \min(H_h H_l, UB(r^{\pi^*}(s,a)) + V_{i+1}(g(s,a))) \tag{1}$$

8:     **for** $s \in S$ **do**
9:         $V_i(s) = \max_{a \in A^h} Q_i(s,a)$
10:     **for** time step $h = 1, ..., H_h$ **do**
11:         Take greedy high-level action $a_h^k = \arg\max_{a \in A^h} Q_h(s_h^k, a)$
12:         Traverse sub-MDP $M(s_h^k, a_h^k)$ with current sub-policy $\pi_{s_h^k, a_h^k}^{N^{k,h}}$ and transition to $s_{h+1}^k$,
        human supervisor provides low-level rewards of the length-$H_l$ roll-out of $\pi_{s_h^k, a_h^k}^{N^{h,k}}$.
13:         Feed low-level rewards into no-regret RL algorithm $\mathcal{A}$ for sub-MDP $M(s_h^k, a_h^k)$. Set the
        sum of the low-level rewards (the intermediate return of $\pi_{s_h^k, a_h^k}^{N^{h,k}}$ in $M(s_h^k, a_h^k)$) as the high-level
        reward $r(s_h^k, a_h^k) = r(\pi_{s_h^k, a_h^k}^{N^{h,k}})$
14:         Add to dataset $D = D \cup \{(h, s_h^k, a_h^k, r(s_h^k, a_h^k)\}$

---

minimization in the cardinal reward case, which as we will see in the ordinal reward case will not always be true.

**Theorem 1** (HRL regret minimization reduces to multi-task, sub-MDP regret minimization). *Let* $UB(\mathcal{R}^{N^{K,H_h}(s,a)})$ *be an upper bound on sub-MDP $M(s,a)$'s cumulative regret over $N^{K,H_h}(s,a)$ visits:*

$$\sum_{k=1}^{K} V_1^{\pi^*}(s_1) - V_1^{\pi^k}(s_1) \leq \tilde{O}\left( \sum_{s,a \in \mathcal{C}(S,A^h)} UB(\mathcal{R}^{N^{K,H_h}(s,a)}) + H^h H^l \sqrt{N^{K,H_h}(s,a)} \right) \tag{2}$$

*Proof Sketch.* We describe the key regret decomposition. After some manipulation, the regret may decompose into the following form, $\sum_{k=1}^{K} V_1^k(s_1) - V_1^{\pi^k}(s_1) \leq \sum_{k=1}^{K} \sum_{h=1}^{H_h} \rho_h^k + \gamma_h^k + \sigma_h^k + \zeta_h^k$, which may be parsed as follows.

$\rho_h^k = UB(r^{\pi^*}(s,a)) - r(\pi_{s_h^k, a_h^k}^{N^{k,h}})$ captures the regret due to sub-optimal intermediate return, the return of $\pi_{S,a}^*$ versus the return of $\pi_{s_h^k, a_h^k}$.

$\gamma_h^k = (P_h - P^{\pi_{k,h}})V_{h+1}^{\pi^*}(s_h^k, a_h^k)$, $\sigma_h^k = (P_h - P^{\pi_{k,h}})(V_{h+1}^k - V_{h+1}^{\pi^*})(s_h^k, a_h^k)$ captures the regret due to sub-optimal policies missing goal-reachability. Here $P_h$ is the idealized transition (goal-reaching), while $P^{\pi_{k,h}}$ is the transition induced by the current sub-policy.

$\zeta_h^k = P^{\pi_{k,h}}(V_{h+1}^k - V_{h+1}^{\pi_k})(s_h^k, a_h^k) - (V_{h+1}^k - V_{h+1}^{\pi_k})(s_{h+1}^k)$ is a martingale difference that concentrates via Azuma Hoeffding, and is dominated by the previous three sums.

Focusing on $\sum_{h=1}^{H_h} \rho_h^k + \gamma_h^k + \sigma_h^k + \zeta_h^k$, we observe that $\gamma_h^k, \sigma_h^k \leq 2H_h H_l P^{\pi_{k,h}}(s_{h+1}^k \neq g(s_h^k, a_h^k))$. The key remaining step is to recognize that $\rho_h^k + \gamma_h^k + \sigma_h^k$ resembles the instantaneous regret in $M(s_h^k, a_h^k)$, and the result follows after some further bounding and rearrangement.

□

For a concrete bound, we note that if $\mathcal{A}$ is set as the classic UCB-VI algorithm, then we attain the usual $\tilde{O}(\sqrt{K})$ regret. Furthermore, we note that our bound is flexible in that one can choose more

specialized learning algorithms $\mathcal{A}$ to leverage prior knowledge. For instance, if it is known that sub-MDPs are linear, one may choose to invoke multi-task RL algorithms that offer more refined regret bounds for $UB(\mathcal{R}^{N^{K,H_h}(s,a)})$ [11].

**Goal Selection:** An astute reader will note that the return of the learned hierarchical policy is close to $V_1^*(s_1)$, the return of the optimal hierarchical policy under *goal function g*. In other words, our learned policy is only as good as the goal function $g$ we choose.

One way to relax the assumption that we have a good goal function $g$ is to assume we have access to multiple goal functions to choose from: $g^1, ..., g^n$. Then, an useful corollary of the sublinear Hier-UCB-VI regret bound, $\frac{1}{K}[\sum_{k=1}^K V_1^{g^i,*}(s_1) - V_1^{g^i,\pi^k}(s_1)] \leq \tilde{O}(\sqrt{K})$, is that it directly implies an UCB on $V_1^{g^i,*}(s_1)$ (optimal return under goal $g^i$). Hence, we may apply any UCB-based bandit algorithm on top of this to compete with the return of the best goal out of all the candidates $\{g^j\}_{j \in [n]}$.

# 4 Learning from Preference Feedback

In the previous section, we develop an algorithm to efficiently learn a hierarchical policy, purely from low-level, cardinal feedback. Now, we consider learning from ordinal (preferences) feedback. Our first observation is that the low-level feedback is no longer sufficient for learning a good policy.

**Proposition 1** (Non-identifiability of ranking among sub-MDP returns). *For any deterministic high-level policy learning algorithm with $N_l$ samples of low-level feedback, there exists a MDP instance that induces regret constant in $N_l$.*

The intuition for this is simply that low-level, ordinal feedback can only identify rankings of low-level policies specific to a goal (sub-MDP), but not necessarily low level policies *across* differing goals. Thus, no matter how large the low-level sample-size $N_l$, the regret is non-vanishing in $N_l$ and hence high-level feedback may be needed to learn. Please see Appendix D for all proofs of results in this section.

## 4.1 Labeler Feedback and Consequences for Reward Modeling

The canonical approach to learning from preferences is reward modeling. Following previous works, we study offline experiment design and assume we have the ability to collect comparison feedback data, in our hierarchical setting both high and low-level data that are then used to learn the reward model [29]. For tractable analysis, we consider the commonly studied linear reward setup [21, 20, 30, 29].

**Assumption 2** (Linear Reward Parametrization). *Suppose we have access to some feature map $\phi : S \times A \to \mathbb{R}^d$, $\mathcal{M}$ has linear reward parametrization w.r.t. $\phi$ if there exists an unknown, reward vector $\theta^* \in \mathbb{R}^d$ such that $r(s,a) = \langle \phi(s,a), \theta^* \rangle$ for all $s, a \in S \times A$.*

Given trajectory $\tau = (s_1, a_1, ..., s_H, a_H)$, we may then define trajectory feature $\phi(\tau) = \sum_{s_i, a_i \in \tau} \phi(s_i, a_i)$, and policy feature expectation under transitions $P$, $\phi^P(\pi) = \mathbb{E}_{\tau \sim \pi, P}[\phi(\tau)]$.

With known feature map $\phi$ and unknown reward parameter $\theta^*$, the preference feedback $o_t$ follows the Bradley-Terry-Luce (BTL) model [6].

**Assumption 3.** *For trajectories $\tau_1, \tau_2$: $\Pr(\tau_1 \succ \tau_2) = \sigma((\theta^*)^T(\phi(\tau_1) - \phi(\tau_2)))$.*

With the definitions out of the way, we now describe a *conceptual challenge* that we encounter when learning from high-level feedback, which as we have shown before may be necessary for learning.

> What can we assume about the high-level labeler's knowledge?

Consider a high level trajectory $\tau_j = \{(s_i^j, a_i^j)\}_{i=1}^{H_h}$. $\phi(\tau_j) = \sum_{i \in [H_h]} \phi(s_i^j, a_i^j)$; the key difficulty is that sub-MDP feature expectation $\phi(s_i^j, a_i^j)$ is dependent on the sub-policy deployed in $M(s_i^j, a_i^j)$. Thus, the high level labeler will have to have in mind some sub-policy $\pi_{s,a}$, when making the comparison. We study two natural types of feedback:

1. **Comparisons based on current sub-policy execution:** It is natural to first assume that the labeler envisions $\phi(s_i^j, a_i^j) = \phi(\pi_{s_i^j, a_i^j}^t)$ at time $t$. In words, it is equivalent to asking: "How well does the high level policy do given *current execution* of sub-goals?"

   *Current-feedback* of this form has the caveat that the labeler will have know about the performance of the current set of sub-policies $\pi_{s,a}^t$ (potentially through AI-assisted means). This knowledge would need to be updated over time as low-level policies $\pi_{s,a}^t$ improve, which introduces a sizable cognitive load.

2. **Comparisons based on idealized sub-policy execution:** To reduce the cognitive load on the labeler, it is natural to fix the sub-policies used in the comparisons. A natural choice then is for the labeler to envision $\phi(s_i^j, a_i^j) = \phi(\pi_{s_i^j, a_i^j}^*)$. In words, it is equivalent to asking: "How well does the high level policy do given *perfect execution* of the sub-goals?" Instantiated in some examples, this would be: "how good is the essay if each argument is fleshed out perfectly" or "how good is the code if each helper function is implemented perfectly".

   *Idealized-feedback* of this form has the caveat that the high-level feedback will be a mismatch of how the current sub-policies actually execute. Although it has the advantage that the labeler is no longer required to (somehow) keep track of low-level sub-policies, thus reducing the cognitive load.

In what follows, we consider both types of feedback, showing that learning from idealized-feedback is possible. As we note, a drawback of idealized-feedback is that it is biased with respect to the realized features (since these are generated under current policies $\pi_{s,a}^t$), while current-feedback is unbiased. We present an upper bound on the bias below.

**Lemma 2** (Bias of idealized-feedback). *Suppose there are $N_h, N_l$ high, low-level trajectories, bias $b$ is such that:* $\|b\|^2 = \sum_{t=1}^{N_h} |\langle \theta^*, \phi^{\pi^{N_l}}(\pi_1^i) - \phi^{\pi^{N_l}}(\pi_2^i) \rangle - \langle \theta^*, \phi^{\pi^*}(\pi_1^i) - \phi^{\pi^*}(\pi_2^i) \rangle|^2 = O(N_h/N_l).$

**Proposition 2** (Reward model learning). *Let $\theta_{MLE} = \arg\min_\theta \ell_D(\theta)$ and let $C_b$ denote an upper bound on bias $C_b \geq \|b\|$, and $\gamma, B$ constants. We have that with high probability:*

$$\|\theta^* - \theta_{MLE}\|_{\hat{\Sigma}_{N^h}^h + \lambda I} \leq C\sqrt{\frac{C_b\sqrt{N_h}}{\gamma^2} + \frac{C_b^2 + d + \log(1/\delta)}{\gamma^2} + \lambda B^2}$$

## 4.2 Hierarchical Preference Learning

We now construct a hierarchical, preference-learning algorithm that invokes REGIME, a contemporary preference-learning algorithm with provable guarantees, as sub-routine for sub-MDP learning [29].

**Sub-MDP reward learning:** To start, we again need to incentivize goal-reaching in the sub-MDP reward. As such, given original feature $\phi_{orig}$, we introduce an additional feature accounting for goal-reachability. For trajectory $\tau$, define $\phi_i(s_i^\tau, a_i^\tau) = [\phi_{orig}(s_i^\tau, a_i^\tau), \mathbb{1}(i = H_l \wedge s_i^\tau = g(s,a))]$ and for policy $\pi$, feature expectation $\phi_i(s_i^\pi, a_i^\pi) = [\phi_{orig}(s_i^\pi, a_i^\pi), \mathbb{1}(i = H_l)\Pr(s_{H_l}^\pi = g(s,a))]$.

The corresponding reward vector will also change to become $\theta^* = [\theta_{orig}^*, \kappa]$ for unknown $\theta_{orig}^*, \kappa$.

**Assumption 4.** *Through instructions to the labeler, $\kappa$ may be raised beyond a threshold of our choosing.*

That is, we assume we can provide instructions to the labeler, emphasizing goal-reachability such that $\kappa$ is higher than some given threshold. As before, we take the threshold to be $\max(C, 2H_h H_l)$. And so while $\kappa$ is unknown, we know that $\kappa \geq \max(C, 2H_h H_l)$. With this set up, we can then bound the regret due to sub-optimal sub-policies, and sub-optimal simulator $P^{\epsilon'}$, both of which are needed in the final regret analysis.

**Lemma 3** (Regret due to sub-optimal sub-policies). *For any high-level policy $\pi$, with high probability:*

$$\langle \phi^{\pi^*, P}(\pi) - \phi^{\pi^{N_l}, P}(\pi), \theta^* \rangle \leq H_h\left(\frac{C_1}{\sqrt{N_l}} + C_2\epsilon'\right)$$

*where this bound makes use of the REGIME guarantee on sub-MDP $M(s,a)$ that $|\langle \phi^P(\pi_{s,a}^*), \theta^* \rangle - \phi^{P^{\epsilon'}}(\pi_{s,a}^{N_l}), \theta^* | \leq \frac{C_1}{\sqrt{N_l}} + C_2\epsilon'$ [29].*

---

**Algorithm 2** Hierarchical-REGIME (Hier-REGIME)

---

**Require:** High-level policy class $\Pi^h$, low level-policy classes $\Pi^l_{s,a}$, simulator $P^{\epsilon'}$ with $\epsilon'$-precision

1: **for** episode $n = 1, ..., N_l$ **do**

2:     $(\pi_1^n, \pi_2^n) \leftarrow \arg\max_{\pi_1, \pi_2 \in \bigcup_{s,a} \Pi^l_{s,a}} \|\phi^{P^{\epsilon'}}(\pi_1) - \phi^{P^{\epsilon'}}(\pi_2)\|_{(\hat{\Sigma}^l_n)^{-1}}$     $\triangleright$ *explore using policy feature expectation across sub-MDPs*

3:     $\hat{\Sigma}^l_{n+1} = \hat{\Sigma}^l_n + (\phi^{P^{\epsilon'}}(\pi_1^n) - \phi^{P^{\epsilon'}}(\pi_2^n))(\phi^{P^{\epsilon'}}(\pi_1^n) - \phi^{P^{\epsilon'}}(\pi_2^n))^T$

4:     Generate trajectories $\tau_1^n, \tau_2^n$ and acquire comparison feedback $o_n$ $\triangleright$ *comparison feedback for the pair of length-$H_l$ trajectories*

5: Compute MLE $\hat{\theta}^l$ from $\{\tau_1^n, \tau_2^n\}_{n=1}^{N_l}$ and $\{o_n\}_{n=1}^{N_l}$

6: Compute $\pi^{N_l}_{s,a} = \arg\max_{\pi \in \Pi^l_{s,a}} \langle \phi^{P^{\epsilon'}}(\pi), \hat{\theta}^l \rangle$

7: **for** episode $n = 1, ..., N_h$ **do**

8:     $(\pi_1^n, \pi_2^n) \leftarrow \arg\max_{\pi_1, \pi_2 \in \Pi^h} \|\phi^{\pi^{N_l}, P^{\epsilon'}}(\pi_1) - \phi^{\pi^{N_l}, P^{\epsilon'}}(\pi_2)\|_{(\hat{\Sigma}^h_n)^{-1}}$     $\triangleright$ *high-level policy feature expectation generated using $\pi^{N_l}_{s,a}$*

9:     $\hat{\Sigma}^h_{n+1} = \hat{\Sigma}^h_n + (\phi^{\pi^{N_l}, P^{\epsilon'}}(\pi_1) - \phi^{\pi^{N_l}, P^{\epsilon'}}(\pi_2))(\phi^{\pi^{N_l}, P^{\epsilon'}}(\pi_1) - \phi^{\pi^{N_l}, P^{\epsilon'}}(\pi_2))^T$

10:     Generate trajectories $\tau_1^n, \tau_2^n$ and acquire comparison feedback $o_n$ $\triangleright$ *comparison feedback for the pair of length-$H_h$ trajectories*

11: Compute MLE $\hat{\theta}^h$ from $\{\tau_1^i, \tau_2^i\}_{i=1}^{N_h}$ and $\{o_i\}_{i=1}^{N_h}$

12: **return** high-level policy $\hat{\pi} = \arg\max_{\pi \in \Pi^h} \langle \phi^{\pi^{N_l}, P^{\epsilon'}}(\pi), \hat{\theta}^h \rangle$, low-level policies $\{\pi^{N_l}_{s,a}\}_{s,a \in S \times A^h}$

---

**Lemma 4** (Regret due to sub-optimal simulator $P^{\epsilon'}$). *Let $\Phi^{\pi^{N_l}, P^{\epsilon'}}(\pi)$ denote the feature expectation under high level policy $\pi$, sub-MDP policies $\pi^{N_l}$ and transitions $P^{\epsilon'}$. With high probability, for any high level policy $\pi$:*

$$|\langle \phi^{\pi^{N_l}, P}(\pi) - \phi^{\pi^{N_l}, P^{\epsilon'}}(\pi), \theta^* \rangle| \leq O((H_h d^2 + H_h^3 H_l^2)\epsilon' + \frac{H_h^2 H_l}{\kappa})$$

### 4.3 Hier-REGIME Analysis

Now, we present the Hier-REGIME Algorithm 2. At a high-level description goes as follows. First, we invoke one copy of REGIME across all sub-MDPs with shared exploration (L1-4) and learned reward (L5). Next, we use the learned reward to compute sub-MDP policies $\pi^{N_l}_{s,a}$ for each sub-MDP $M(s, a)$ (L6). Finally, we invoke one copy of REGIME for the high-level MDP, where the feature function is defined as $\phi^{\pi^{N_l}_{s,a}, P^{\epsilon'}}$ (L8). Next, we note two properties about Algorithm 2.

**Hierarchical Exploration:** A key aspect of experiment design in offline RL is ensuring sufficient coverage with exploration. The difficulty with coverage in the hierarchical setting is that at first glance, we may need to search for pairs of trajectories over $(\pi_1, \{\pi^1_{s,a}\}), (\pi_1, \{\pi^2_{s,a}\}) \in (\Pi^h, \bigtimes_{s,a} \Pi^l_{s,a})$, instead of over $\pi_1, \pi_2 \in \Pi^h$. However, we show that in the goal-HRL case, we can fix the sub-policies to be $\pi^{N_l}_{s,a}$ (for $N_l$ large enough), and this is sufficient to compete with the optimal, hierarchical policy.

Additionally, unlike the tabular setting, sub-MDPs now share a common reward parameter $\theta^*$, thus allowing us to jointly, instead of separately as in tabular case, explore across sub-MDPs.

**Sufficiency of low-level feedback:** Through the algorithm, we can observe that low- and high-level exploration generates feature expectations set: $\{\phi^{P^{\epsilon'}}(\pi_1) - \phi^{P^{\epsilon'}}(\pi_2) \mid \pi_1, \pi_2 \in \bigcup_{s,a} \Pi^l_{s,a}\}$ and $\{\phi^{P^{\epsilon'}}(\pi_1) - \phi^{P^{\epsilon'}}(\pi_2) \mid \pi_1, \pi_2 \in \Pi^h, \pi_{s,a} = \pi^{N_l}_{s,a} \; \forall s, a\}$. Therefore, when coverage of high level policy is subsumed by low-level features already (the latter is a subset of the former), it suffices to explore only using low-level feedback. As shown before in Proposition 2, it is not always sufficient. However, as we will see below, when it is sufficient, using low-level feedback leads to better rates. First, we derive the regret decomposition and then use it evaluate the sample complexity.

**Theorem 2.** *With high probability, under $N_h > 0$:*

$$V^{\pi^*, \pi^*} - V^{\hat{\pi}, \pi^{N_l}}$$

$$\leq \langle \phi^{\pi^*, P}(\pi^*) - \phi^{\pi^{N_l}, P}(\pi^*), \theta^* \rangle + \frac{1}{\sqrt{N_h}}(2d \log(1 + \frac{N_h}{d}))\|\theta^* - \hat{\theta}\|_{\hat{\Sigma}^h_{N_h}} +$$

$$|\langle \phi^{\pi^{N_l}, P}(\pi^*) - \phi^{\pi^{N_l}, P^{\epsilon'}}(\pi^*), \theta^* \rangle| + |\langle \phi^{\pi^{N_l}, P^{\epsilon'}}(\hat{\pi}) - \phi^{\pi^{N_l}, P}(\hat{\pi}), \theta^* \rangle|$$

To parse this, the regret decomposes into four terms. The first term is the regret due to sub-optimality in low-level policies $\pi^{N_l}$. The remaining three terms are derived from sub-optimality due to high-level policy $\hat{\pi}$, decomposing into the second term on regret due to bias in learned reward $\hat{\theta}$, the third and fourth term on regret due to sub-optimality of simulator $P^{\epsilon'}$.

A main benefit of developing a learning Algorithm 2 is that we can then quantitatively assess the sample complexity associated with the two types of human feedback. As one may expect, there is a tradeoff between better sample complexity and cognitive load, with current-feedback attaining better sample efficiency but also requiring higher cognitive load on the human supervisor.

**Corollary 1.** *Using Theorem 2, we obtain the following rates in terms of data tradeoffs:*

- ***Idealized-feedback and required high-/low-level feedback:*** *the overall rate comes out to $O(N_l^{-1/4} + N_h^{-1/2})$. While high level trajectories provide additional coverage, it also incurs bias linear in $N_h$ of the bias of the low-level trajectories, thus slowing down the rate (Lemma 2).*

- ***Current-feedback and required high-/low-level feedback:*** *the overall rate comes out to $O(N_l^{-1/2} + N_h^{-1/2})$. The current-feedback is unbiased and results in more efficient reward learning with $\|\theta^* - \hat{\theta}\|_{\hat{\Sigma}^h_{N_h}} = O(1)$ [29].*

- ***Only low-level feedback is required due to sufficiency in coverage:*** *the overall rate comes out to $O(N_l^{-1/2})$. In a nutshell, this is because we can explore with just $N_l$ low-level samples which is unbiased, resulting in $\|\theta^* - \hat{\theta}\|_{\hat{\Sigma}^l_{N_l}} = O(1)$. Hence, both exploration and reward learning is efficient.*

## 5 Discussion

Our work considers scalable oversight in the context of goal-conditioned HRL, in which we show that one can efficiently use hierarchical structure to learn from bounded human feedback.

**Limitations & Future Work:** In goal-conditioned HRL, our regret guarantees are with respect to the return of the optimal, hierarchical policy, whose performance is dependent on the usefulness of goal function $g$. Further research is needed to understand on how to learn good goal functions, using limited supervised or unsupervised learning. Additionally, under current-feedback, the labeler providing high-level feedback is somehow made aware of sub-policy performance. An exciting research direction is how one may provide such knowledge through AI-assistance.

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

# A  More Related Works

**Scalable Oversight:** Scalable oversight is a nascent but important topic in the area of AI alignment [2, 8, 15, 5], wherein the goal is to boost the labeler's ability to provide feedback to complex models. Proposed approaches include (recursive) self-critique, summarization, debate, plain model Interaction and market-making, all of which aim to have the model (or auxiliary models) generate interpretable and/or lower-dimensional forms of outputs for the human to parse [15, 13, 24, 27, 23, 5, 12]. Our work studies how one may leverage hierarchical structure as one approach to scaling up feedback.

**Goal-conditioned RL:** Further afield, there has been a lot of work demonstrating the promise/success of goal-conditioned RL with examples from the likes of [16, 17, 7, 10]. The sub-MDP reward is often set to incentivize *only* goal state reachability, as oftentimes the MDP of interest has sparse rewards, making intermediate returns zero. In our setting, rewards need not be sparse, thus bringing into consideration the tradeoff between intermediate return and goal-reachability. This work initiates the study of scalable oversight in goal-oriented HRL, and owing to the success of goal-oriented HRL in practice, it is our hope that it can be stepping stone towards developing practical scalable oversight techniques.

# B  Concrete Hierarchical MDP Example

The prototypical example in HRL is the maze, as studied in for instance [17, 22]. A maze consists of rooms with doors. The goal is to get to the exit in as few steps as possible. The MDP may be defined as follows:

- For the global MDP, $S = S^h \times S^l$ where $s^h$ denotes the index of the current room, and $s^l$ denotes the position of the agent in the room. Action set $A$ consists of moving (L, R, U, D, Stay).
- For the High-level MDP, high-level action $A^h$ consists of moving to the (N, S, E, W) door of the room. $s$ is the current location of the agent, and $g(s, a^h)$ maps the goal (door) to its location.
- For the Low-level MDP, it has state space $S^l_{s,a} \subset S$ and the action set $A$ is the same moving (U, D, L, R, Stay).

As noted in the previous section, HRL algorithms can achieve superior statistical sample complexity when there is lots of repeated sub-MDP structure (there are many isomoprhic rooms) and each room has small state-space size [25].

| Notation | |
|---|---|
| $M(s,a)$ | sub-MDP at state $s$ with high level action $a$ |
| $\pi_{s,a}^i$ | policy used by sub-MDP $M(s,a)$'s no-regret algorithm during the $i$-th visit |
| $\pi_{s,a}^*$ | optimal policy in sub-MDP $M(s,a)$ |
| $r(\pi_{s,a}^i)$ | expected reward of policy $\pi_{s,a}^i$ in sub-MDP $M(s,a)$ |
| $r_{l,h}$ | sub-MDP reward definition. |
| $\hat{r}(\pi_{s,a}^i)$ | observed reward of policy $\pi$ in sub-MDP $M(s,a)$ |
| $\bar{r}_n(s,a)$ | average observed policy reward $\bar{r}_n(s,a) = \frac{1}{n}\sum_{i=1}^n \hat{r}(\pi_{s,a}^i)$ |
| $\mathcal{R}^n(s,a)$ | sub-MDP $M(s,a)$ cumulative regret across $n$ steps, $\mathcal{R}^n(s,a) = \sum_{i=1}^n r(\pi_{s,a}^*) - r(\pi_{s,a}^i)$ |
| $N^{k,h}(s,a)$ | number of times $M(s,a)$ has been visited up until episode $k$, horizon $h$ |
| $P^\pi(\cdot \mid s,a)$ | distribution over states of policy $\pi$ after going through subMDP $M(s,a)$ |
| $\psi_n$ | a factor such that $\psi_n = \tilde{O}(\sqrt{n})$, where the $\tilde{O}$ omits up to log dependence on $K$ |

Table 1: Table of notation used in this section.

## C  Proofs for Section 3

### C.1  Sub-MDP Bonus Construction

**Sub-MDP Reward Definition:** Define the reward in sub-MDP $M(s,a)$ at time step $h$ to be:
$r_{l,h}(s',a') = r(s',a') + \kappa \mathbb{1}(h = H_l \wedge s' = g(s,a))$.

Firstly, since by definition $\pi_{s,a}^* \in \arg\max_{\pi \in \Pi_{s,a}} r(\pi) + C \cdot \Pr(s_{H_l}^\pi = g(s,a))$, we have that
$\pi_{s,a}^* \in \arg\max_{\pi \in \Pi_{s,a}} r(\pi) + \kappa \cdot \Pr(s_{H_l}^\pi = g(s,a))$.

Indeed,

$$r(\pi_{s,a}^*) + \kappa \Pr(s_{H_l}^{\pi_{s,a}^*} = g(s,a))$$
$$= [r(\pi_{s,a}^*) + C \cdot \Pr(s_{H_l}^{\pi_{s,a}^*} = g(s,a))] + (\kappa - C)\Pr(s_{H_l}^{\pi_{s,a}^*} = g(s,a))$$
$$\geq [r(\pi) + C \cdot \Pr(s_{H_l}^\pi = g(s,a))] + (\kappa - C)\Pr(s_{H_l}^\pi = g(s,a))$$
$$(\Pr(s_{H_l}^{\pi_{s,a}^*} = g(s,a)) = 1 \geq \Pr(s_{H_l}^\pi = g(s,a)) \,\forall \pi)$$

Secondly, using the definition of $r_l$, we have that:

$$r_l(\pi_{s,a}^*) - r_l(\pi_{s,a}^i) = r(\pi_{s,a}^*) + \kappa P(s_{H_l}^{\pi_{s,a}^*} = g(s,a)) - r(\pi_{s,a}^i) - \kappa P(s_{H_l}^{\pi_{s,a}^i} = g(s,a))$$

By the reachability assumption, $P(s_{H_l}^{\pi_{s,a}^*} = g(s,a)) = 1$, this implies that

$$r_l(\pi_{s,a}^*) - r_l(\pi_{s,a}^i) = r(\pi_{s,a}^*) - r(\pi_{s,a}^i) + \kappa P(s_{H_l}^{\pi_{s,a}^i} \neq g(s,a))$$

Therefore, summing this across $n$ visits to $M(s,a)$, we have:

$$\mathcal{R}^n(s,a)$$
$$= \sum_{i=1}^n r_l(\pi_{s,a}^*) - r_l(\pi_{s,a}^i)$$
$$= \sum_{i=1}^n r(\pi_{s,a}^*) - r(\pi_{s,a}^i) + \kappa \sum_{i=1}^n P(s_{H_l}^{\pi_{s,a}^i} \neq g(s,a))$$

This statement is useful because we can compute an UCB on $\sum_{i=1}^n r(\pi_{s,a}^*)$ and, implicitly, a LCB on $\sum_{i=1}^n r(\pi_{s,a}^i)$ (provided we do not bound $\mathcal{R}^n(s,a)$).

**Lemma 5** (Bonus with "penalty" for non-reachability). *Let $UB(\mathcal{R}^n(s,a))$ be any upper bound on the sub-MDP regret, then if we define:*

$$b_r^{s,a}(n) = \frac{UB(\mathcal{R}^n(s,a)) + (\kappa + H_l)2\log(\frac{|\mathcal{C}(S,A^h)|H_h K}{\delta})\sqrt{n}}{n} - \frac{\kappa}{n}\sum_{i=1}^{n} \mathbb{1}(s_{H_l}^{\pi_{s,a}^i} \neq g(s,a))$$

*Then, $\bar{r}_n(s,a) + b_r^{s,a}(n)$ is an UCB for $r(\pi_{s,a}^*)$ with probability $\geq 1 - \frac{\delta}{3|\mathcal{C}(S,A^h)|H_h K}$.*

*Let the event that the above holds be $\mathcal{E}_{s,a}^n$.*

*Proof.*

$$\sum_{i=1}^{n} r(\pi_{s,a}^*)$$

$$= \mathcal{R}^n(s,a) - \kappa\sum_{i=1}^{n} P(s_{H_l}^{\pi_{s,a}^i} \neq g(s,a)) + \sum_{i=1}^{n} r(\pi_{s,a}^i)$$

$$\leq \mathcal{R}^n(s,a) - \kappa\left(\sum_{i=1}^{n} \mathbb{1}(s_{H_l}^{\pi_{s,a}^i} \neq g(s,a)) - \psi_n\right) + \sum_{i=1}^{n} r(\pi_{s,a}^i) \qquad (\diamond)$$

$$= \mathcal{R}^n(s,a) - \kappa\sum_{i=1}^{n} \mathbb{1}(s_{H_l}^{\pi_{s,a}^i} \neq g(s,a)) + \kappa\psi_n + \sum_{i=1}^{n} \hat{r}(\pi_{s,a}^i) + \left(\sum_{i=1}^{n} r(\pi_{s,a}^i) - \sum_{i=1}^{n} \hat{r}(\pi_{s,a}^i)\right)$$

$$\leq UB(\mathcal{R}^n(s,a)) + (\kappa + H_l)\psi_n - \kappa\sum_{i=1}^{n} \mathbb{1}(s_{H_l}^{\pi_{s,a}^i} \neq g(s,a)) + \sum_{i=1}^{n} \hat{r}(\pi_{s,a}^i) \qquad (\kappa' = \kappa + H_l)$$

$(\diamond)$ : Here we use two applications of Azuma-Hoeffding:

- With probability higher than $1 - \delta$:

$$|\sum_{i=1}^{n} P(s_{H_l}^{\pi_{s,a}^i} \neq g(s,a)) - \sum_{i=1}^{n} \mathbb{1}(s_{H_l}^{\pi_{s,a}^i} \neq g(s,a))| \leq \psi_n = 2\sqrt{n}$$

We have that $\mathbb{E}[P(s_{H_l}^{\pi_{s,a}^i} \neq g(s,a)) - \mathbb{1}(s_{H_l}^{\pi_{s,a}^i} \neq g(s,a))|\mathcal{F}_{i-1}] = 0$.

This is true because $P(s_{H_l}^{\pi_{s,a}^i} \neq g(s,a))$ and $\mathbb{1}(s_{H_l}^{\pi_{s,a}^i} \neq g(s,a)$ are a function of only the transition probability of the MDP at the $i$th step conditioned on $\mathcal{F}_{i-1}$. Thus, $P(s_{H_l}^{\pi_{s,a}^i} \neq g(s,a)) - \mathbb{1}(s_{H_l}^{\pi_{s,a}^i} \neq g(s,a))$ is a martingale difference. And we can use Azuma-Hoeffding.

- With probability higher than $1 - \delta$:

$$|\sum_{i=1}^{n} r(\pi_{s,a}^i) - \sum_{i=1}^{n} \hat{r}(\pi_{s,a}^i)| \leq H_l\psi_n \leq H_l 2\sqrt{n}$$

This again follows from Azuma-Hoeffding on martingale difference $r(\pi_{s,a}^i) - \hat{r}(\pi_{s,a}^i)$, as $\mathbb{E}[r(\pi_{s,a}^i) - \hat{r}(\pi_{s,a}^i)|\mathcal{F}_{i-1}] = 0$. And $|r(\pi_{s,a}^i) - \hat{r}(\pi_{s,a}^i)| \leq H_l$.

Thus,

$$r(\pi_{s,a}^*) \leq \frac{1}{n}\sum_{i=1}^{n} \hat{r}(\pi_{s,a}^i) + b_r^{s,a}(n) \Rightarrow r(\pi_{s,a}^*) - \bar{r}_n(s,a) \leq b_r^{s,a}(n)$$

$\square$

**Remark 1.** *One choice for $UB(\mathcal{R}^n(s,a)) = H_l^{3/2}\sqrt{|S_{s,a}^l||A|n}$ if we let $\mathcal{A}_{s,a}$ be the standard UCB-VI algorithm [3].*

## C.2 Optimism Lemma

**Lemma 6** (Optimism). *Let $V_h^k$ be the V value as in Algorithm 1 at episode $k$. Let $\pi^*$ be the optimal hierarchical policy. For a fixed $k$ and $h$, if $\forall s,a,n$, $\mathcal{E}_{s,a}^n$ holds, then:*

$$V_h^k(s) \geq V_h^{\pi^*}(s) \quad \forall s$$

*Proof.* Fix some episode $k$. We will prove this lemma via induction on $h = H_h + 1, ..., 1$.

**Base case:** At $h = H_h + 1$, $V_h^k(s) \geq 0 = V_h^{\pi^*}(s)$ for all $s$.

**Induction Step:** Suppose this is true for up until $h = H_h + 1, ..., h' + 1$. Now at time step $h'$ and any $s, a$.

Firstly, if $Q_{h'}^k(s,a) = H_h H_l$ (e.g. if $s, a \notin \mathcal{D}^k$), then $Q_{h'}^k(s,a) \geq Q_{h'}^*(s,a)$. Otherwise, $Q_{h'}^k(s,a) < H_h H_l$ and we have that:

$$Q_{h'}^k(s,a) - Q_{h'}^*(s,a) = [\bar{r}_{N^{k,h}(s,a)}(s,a) + b_r^{s,a}(N^{k,h}(s,a)) + V_{h'+1}^k(g(s,a))] - (r(\pi_{s,a}^*) + P_{h'}V_{h'+1}^{\pi^*}(s,a))$$
$$(Q_{h'}^k \text{ definition as in Equation 1})$$

$$\geq V_{h'+1}^k(g(s,a)) - P_{h'}V_{h'+1}^{\pi^*}(s,a)$$
$$(\bar{r}_{N^{k,h}(s,a)}(s,a) + b_r^{s,a}(N^{k,h}(s,a)) \text{ is an UCB of } r(\pi_{s,a}^*))$$

$$= V_{h'+1}^k(g(s,a)) - V_{h'+1}^{\pi^*}(g(s,a))$$
$$(\pi_{s,a}^* \text{ reaches goal state w.p 1, so } P_{h'}(g(s,a)|s,a) = 1)$$

$$\geq 0 \qquad\qquad\qquad\qquad (\text{induction hypothesis})$$

Thus, $V_{h'}^k(s) = \max_a Q_{h'}^k(s,a) \geq \max_a Q_{h'}^*(s,a) = V_{h'}^{\pi^*}(s)$.

$\square$

**Corollary 2.**

$$\sum_{k=1}^K V_1^{\pi^*}(s_1) - V_1^{\pi^k}(s_1) \leq \sum_{k=1}^K V_1^k(s_1) - V_1^{\pi^k}(s_1)$$

## C.3 Supporting results needed for regret analysis

**Proposition 3.**
$$\sum_{k=1}^K V_1^k(s_1) - V_1^{\pi^k}(s_1) \leq \sum_{k=1}^K \sum_{h=1}^{H_h} \zeta_h^k + \gamma_h^k + \sigma_h^k + \rho_h^k \tag{3}$$

*Proof.* For any $k$ and $h$, we consider bounding $V_h^k(s_h^k) - V_h^{\pi_k}(s_h^k)$, which is equal to:

$$V_h^k(s_h^k) - V_h^{\pi_k}(s_h^k) = (Q_h^k - Q_h^{\pi_k})(s_h^k, a_h^k)$$
$$\leq (\bar{r}_{N^{k,h}(s_h^k,a_h^k)}(s_h^k, a_h^k) + b_r^{s_h^k,a_h^k}(N^{k,h}(s_h^k,a_h^k))) - r(\pi_{s_h^k,a_h^k}^{N^{k,h}(s_h^k,a_h^k)})$$
$$+ V_{h+1}^k(g(s_h^k, a_h^k)) - P^{\pi_{k,h}}V_{h+1}^{\pi_k}(s_h^k, a_h^k) \qquad (\text{due to the } \min)$$
$$= \rho_h^k + [V_{h+1}^k(g(s_h^k, a_h^k)) - P^{\pi_{k,h}}V_{h+1}^{\pi_k}(s_h^k, a_h^k)]$$

where we set $\rho_h^k = \bar{r}_{N^{k,h}(s_h^k,a_h^k)}(s_h^k, a_h^k) + b_r^{s_h^k,a_h^k}(N^{k,h}(s_h^k,a_h^k))) - r(\pi_{s_h^k,a_h^k}^{N^{k,h}(s_h^k,a_h^k)})$.

Continuing with the original proof and focusing on the second term:

$$V_{h+1}^k(g(s_h^k, a_h^k)) - P^{\pi_{k,h}} V_{h+1}^{\pi_k}(s_h^k, a_h^k)$$
$$= V_{h+1}^k(g(s_h^k, a_h^k)) - P^{\pi_{k,h}} V_{h+1}^k(s_h^k, a_h^k) + P^{\pi_{k,h}}(V_{h+1}^k - V_{h+1}^{\pi_k})(s_h^k, a_h^k)$$
$$= (P_h - P^{\pi_{k,h}}) V_{h+1}^k(s_h^k, a_h^k) + P^{\pi_{k,h}}(V_{h+1}^k - V_{h+1}^{\pi_k})(s_h^k, a_h^k)$$

($P^h$ is the transition under optimal sub MDP policy so it takes $s_h^k, a_h^k$ to $g(s_h^k, a_h^k)$ deterministically)

$$= (P_h - P^{\pi_{k,h}}) V_{h+1}^{\pi^*}(s_h^k, a_h^k) + (P_h - P^{\pi_{k,h}})(V_{h+1}^k - V_{h+1}^{\pi^*})(s_h^k, a_h^k) + P^{\pi_{k,h}}(V_{h+1}^k - V_{h+1}^{\pi_k})(s_h^k, a_h^k)$$
$$= \gamma_h^k + \sigma_h^k + P^{\pi_{k,h}}(V_{h+1}^k - V_{h+1}^{\pi_k})(s_h^k, a_h^k)$$

where

- $\gamma_h^k = (P_h - P^{\pi_{k,h}}) V_{h+1}^{\pi^*}(s_h^k, a_h^k)$

- $\sigma_h^k = (P_h - P^{\pi_{k,h}})(V_{h+1}^k - V_{h+1}^{\pi^*})(s_h^k, a_h^k)$

In summary,

$$V_h^k(s_h^k) - V_h^{\pi_k}(s_h^k)$$
$$\le \rho_h^k + \gamma_h^k + \sigma_h^k + P^{\pi_{k,h}}(V_{h+1}^k - V_{h+1}^{\pi_k})(s_h^k, a_h^k)$$
$$= (V_{h+1}^k - V_{h+1}^{\pi_k})(s_{h+1}^k) + \zeta_h^k + \gamma_h^k + \sigma_h^k + \rho_h^k,$$

where we introduce the notation $\zeta_h^k = P^{\pi_{k,h}}(V_{h+1}^k - V_{h+1}^{\pi_k})(s_h^k, a_h^k) - (V_{h+1}^k - V_{h+1}^{\pi_k})(s_{h+1}^k)$.
Unrolling the recursion starting at $h = 1$:

$$V_1^k(s_h^k) - V_1^{\pi_k}(s_h^k)$$
$$\le 1(\zeta_h^k + \gamma_h^k + \sigma_h^k + \rho_h^k) + ... + (1)^{H_h}(\zeta_{H_h}^k + \gamma_{H_h}^k + \sigma_{H_h}^k + \rho_{H_h}^k)$$
$$= 1 \cdot \left( \sum_{h=1}^{H_h} \zeta_h^k + \gamma_h^k + \sigma_h^k + \rho_h^k \right)$$

Summing across $k \in [K]$, it suffices to bound:

$$\sum_{k=1}^{K} V_1^k(s_1) - V_1^{\pi^k}(s_1) \le \sum_{k=1}^{K} \sum_{h=1}^{H_h} \zeta_h^k + \gamma_h^k + \sigma_h^k + \rho_h^k \tag{4}$$

$\square$

**Remark 2.** *There are two sources of sub-optimality in the bound.*

*One is the sub-optimality while executing the sub-MDP policies. This is covered by the per-step high level reward bonus (which is also the UCB on the return of the sub-MDP's return) in $\rho_h^k$.*

*The other is the sub-optimality of not landing on $g(s_h^k, a_h^k)$, there is covered by $\gamma_h^k, \sigma_h^k$, which affects future reward. The martingale difference $\zeta_h^k$ is zero in expectation, so it is not some measure of suboptimality.*

We first bound the $\zeta$'s, whose sum is dominated by $\sum_{k=1}^{K} \sum_{h=1}^{H_h} \rho_h^k + \gamma_h^k + \sigma_h^k$.

**Lemma 7.** *With probability $\geq 1 - \delta/3$:*

$$\sum_{k=1}^{K}\sum_{h=1}^{H_h}\zeta_h^k \leq \tilde{O}(H^h H^l \sqrt{H^h K})$$

*Let the event that the above inequality hold be $\mathcal{E}^\zeta$.*

*Proof.* The concentration of $\zeta_h^k$ follows from Azuma Hoeffding, as the following is a martingale difference.

$$\zeta_h^k = P^{\pi_{k,h}}(V_{h+1}^k - V_{h+1}^{\pi_k})(s_h^k, a_h^k) - (V_{h+1}^k - V_{h+1}^{\pi_k})(s_{h+1}^k)$$

with $\mathbb{E}[\zeta_h^k | F_{k,h}] = 0$, since the expectation is only wrt randomness in $s_{h+1}^k$. Moreover, this martingale difference is bounded by $4H^h H^l$

$\square$

Next, we simplify the sum of remaining terms.

**Lemma 8.** *We have that:*

$$\sum_{k=1}^{K}\sum_{h=1}^{H_h}\gamma_h^k \leq H^h H^l \sum_{k=1}^{K}\sum_{h=1}^{H_h} P^{\pi_{k,h}}(s_{h+1}^k \neq g(s_h^k, a_h^k))$$

*and*

$$\sum_{k=1}^{K}\sum_{h=1}^{H_h}\sigma_h^k \leq H^h H^l \sum_{k=1}^{K}\sum_{h=1}^{H_h} P^{\pi_{k,h}}(s_{h+1}^k \neq g(s_h^k, a_h^k))$$

*Proof.*

$$\sum_{k=1}^{K}\sum_{h=1}^{H_h}\gamma_h^k$$

$$= \sum_{k=1}^{K}\sum_{h=1}^{H_h}(P_h - P^{\pi_{k,h}})V_{h+1}^{\pi^*}(s_h^k, a_h^k)$$

$$= \sum_{k=1}^{K}\sum_{h=1}^{H_h}P^{\pi_{k,h}}(s_{h+1}^k \neq g(s_h^k, a_h^k))(V_{h+1}^{\pi^*}(g(s_h^k, a_h^k)) - V_{h+1}^{\pi^*}(s_{h+1}^k))$$

$$\leq H^h H^l \sum_{k=1}^{K}\sum_{h=1}^{H_h}P^{\pi_{k,h}}(s_{h+1}^k \neq g(s_h^k, a_h^k))$$

Similarly,

$$\sum_{k=1}^{K}\sum_{h=1}^{H_h}\sigma_h^k$$

$$=\sum_{k=1}^{K}\sum_{h=1}^{H_h}(P_h-P^{\pi_{k,h}})(V_{h+1}^k-V_{h+1}^{\pi^*})(s_h^k,a_h^k)$$

$$=\sum_{k=1}^{K}\sum_{h=1}^{H_h}P^{\pi_{k,h}}(s_{h+1}^k\neq g(s_h^k,a_h^k))[(V_{h+1}^k-V_{h+1}^{\pi^*})(g(s_h^k,a_h^k))-(V_{h+1}^k-V_{h+1}^{\pi^*})(s_{h+1}^k)]$$

$$\leq H^h H^l\sum_{k=1}^{K}\sum_{h=1}^{H_h}P^{\pi_{k,h}}(s_{h+1}^k\neq g(s_h^k,a_h^k))$$

$\square$

**Lemma 9.** *With probability* $\geq 1-\delta/3$:

$$\sum_{k=1}^{K}\sum_{h=1}^{H_h}\rho_h^k\leq\sum_{s,a\in\mathcal{C}(S,A^h)}\sum_{i=1}^{N^{K,H_h}(s,a)}r(\pi_{s,a}^*)-r(\pi_{s,a}^i)+\sum_{s,a\in\mathcal{C}(S,A^h)}\sum_{i=1}^{N^{K,H_h}(s,a)}\frac{UB(\mathcal{R}^i(s,a))-\mathcal{R}^i(s,a)+(\kappa''+\kappa)\psi_i}{i}$$

*Let $\mathcal{E}^\rho$ be the event that this holds.*

*Proof.* We first expand the $\rho_h^k$ sum:

$$\sum_{k=1}^{K}\sum_{h=1}^{H_h}\rho_h^k$$

$$=\sum_{k=1}^{K}\sum_{h=1}^{H_h}\bar{r}_{N^{k,h}(s_h^k,a_h^k)}(s_h^k,a_h^k)+b_r^{s_h^k,a_h^k}(N^{k,h}(s_h^k,a_h^k))-r(\pi_{s_h^k,a_h^k}^{N^{k,h}(s_h^k,a_h^k)})$$

$$=\sum_{s,a\in\mathcal{C}(S,A^h)}\sum_{i=1}^{N^{K,H_h}(s,a)}\bar{r}_i(s,a)+b_r^{s,a}(i)-r(\pi_{s,a}^i)$$

$$=\sum_{s,a\in\mathcal{C}(S,A^h)}\sum_{i=1}^{N^{K,H_h}(s,a)}\frac{1}{i}\sum_{j=1}^{i}\hat{r}(\pi_{s,a}^j)+\frac{UB(\mathcal{R}^i(s,a))+\kappa'\psi_i-\kappa\sum_{j=1}^{i}\mathbb{1}(s_{H_l}^{\pi_{s,a}^j}\neq g(s,a))}{i}-r(\pi_{s,a}^i)$$

$$\text{(using definition of bonus)}$$

$$\leq\sum_{s,a\in\mathcal{C}(S,A^h)}\sum_{i=1}^{N^{K,H_h}(s,a)}\frac{1}{i}\sum_{j=1}^{i}r(\pi_{s,a}^j)+\frac{H_l\psi_i}{i}+\frac{UB(\mathcal{R}^i(s,a))+\kappa'\psi_i-\kappa\sum_{j=1}^{i}\mathbb{1}(s_{H_l}^{\pi_{s,a}^j}\neq g(s,a))}{i}-r(\pi_{s,a}^i)$$

$$\text{(Azume-Hoeffding for concentration of }\hat{r}\text{ around }r\text{)}$$

Using the two-sided concentration bound we had before (the other way): $\sum_{j=1}^{i}\mathbb{1}(s_{H_l}^{\pi_{s,a}^j}\neq g(s,a))+\psi_i\geq\sum_{j=1}^{i}P(s_{H_l}^{\pi_{s,a}^j}\neq g(s,a))$ w.h.p:

$$\sum_{j=1}^{i}r(\pi_{s,a}^*)-r(\pi_{s,a}^j)\geq\mathcal{R}^i(s,a)-\kappa(\sum_{j=1}^{i}\mathbb{1}(s_{H_l}^{\pi_{s,a}^j}\neq g(s,a))+\psi_i)$$

$$\Rightarrow\sum_{j=1}^{i}r(\pi_{s,a}^*)-\mathcal{R}^i(s,a)+\kappa\psi_i\geq\sum_{j=1}^{i}r(\pi_{s,a}^j)-\kappa\sum_{j=1}^{i}\mathbb{1}(s_{H_l}^{\pi_{s,a}^j}\neq g(s,a))$$

We continue our derivation:

$$\sum_{s,a\in\mathcal{C}(S,A^h)} \sum_{i=1}^{N^{K,H_h}(s,a)} \frac{1}{i}\Big(\sum_{j=1}^{i} r(\pi_{s,a}^j) + UB(\mathcal{R}^i(s,a)) + \kappa''\psi_i - \kappa\sum_{j=1}^{i}\mathbb{1}(s_{H_l}^{\pi_j}\neq g(s,a))\Big) - r(\pi_{s,a}^i)$$

$$(\kappa'' = \kappa' + H_l)$$

$$\leq \sum_{s,a\in\mathcal{C}(S,A^h)} \sum_{i=1}^{N^{K,H_h}(s,a)} \frac{1}{i}\Big[\sum_{j=1}^{i} r(\pi_{s,a}^*) - \mathcal{R}^i(s,a) + \kappa\psi_i\Big] - r(\pi_{s,a}^i) + \sum_{s,a\in\mathcal{C}(S,A^h)} \sum_{i=1}^{N^{K,H_h}(s,a)} \frac{UB(\mathcal{R}^i(s,a)) + \kappa''\psi_i}{i}$$

$$\text{(using the identity above)}$$

$$\leq \sum_{s,a\in\mathcal{C}(S,A^h)} \sum_{i=1}^{N^{K,H_h}(s,a)} r(\pi_{s,a}^*) - r(\pi_{s,a}^i) + \sum_{s,a\in\mathcal{C}(S,A^h)} \sum_{i=1}^{N^{K,H_h}(s,a)} \frac{UB(\mathcal{R}^i(s,a)) - \mathcal{R}^i(s,a) + (\kappa'' + \kappa)\psi_i}{i}$$

$$\square$$

### C.3.1 Overall Regret Bound

**Theorem 3.** *Under events $\bigcap_{s,a,n} \mathcal{E}^n_{s,a} \cap \mathcal{E}^\zeta \cap \mathcal{E}^\rho$, we have that:*

$$\sum_{k=1}^{K}\sum_{h=1}^{H_h} \rho_h^k + \gamma_h^k + \sigma_h^k \leq \sum_{s,a\in\mathcal{C}(S,A^h)} (\log(N^{K,H_h}(s,a))+1)UB(\mathcal{R}^{N^{K,H_h}(s,a)}) + O(H^h H^l \sqrt{N^{K,H_h}(s,a)})$$

*Proof.*

$$\sum_{k=1}^{K}\sum_{h=1}^{H_h} \rho_h^k + \gamma_h^k + \sigma_h^k$$

$$\leq \sum_{s,a\in\mathcal{C}(S,A^h)} \sum_{i=1}^{N^{K,H_h}(s,a)} r(\pi^*_{s,a}) - r(\pi^i_{s,a}) +$$

$$\sum_{s,a\in\mathcal{C}(S,A^h)} \sum_{i=1}^{N^{K,H_h}(s,a)} \frac{UB(\mathcal{R}^i(s,a)) - \mathcal{R}^i(s,a) + \kappa\psi_i}{i} + 2H^h H^l \sum_{k=1}^{K}\sum_{h=1}^{H_h} P^{\pi_{k,h}}(s_{h+1}^k \neq g(s_h^k, a_h^k))$$

$$= \sum_{s,a\in\mathcal{C}(S,A^h)} \sum_{i=1}^{N^{K,H_h}(s,a)} \frac{UB(\mathcal{R}^i(s,a)) - \mathcal{R}^i(s,a) + \kappa\psi_i}{i}$$

$$+ \sum_{s,a\in\mathcal{C}(S,A^h)} \sum_{i=1}^{N^{K,H_h}(s,a)} r(\pi^*_{s,a}) - r(\pi^i_{s,a}) + 2H^h H^l \sum_{s,a\in\mathcal{C}(S,A^h)} \left[ \sum_{i=1}^{N^{K,H_h}(s,a)} P(s_{H_l}^{\pi^i_{s,a}} \neq g(s_h^k, a_h^k)) \right]$$
$$\text{(group third sum by } s, a)$$

$$\leq \sum_{s,a\in\mathcal{C}(S,A^h)} \sum_{i=1}^{N^{K,H_h}(s,a)} \frac{UB(\mathcal{R}^i(s,a)) - \mathcal{R}^i(s,a) + \kappa\psi_i}{i}$$

$$+ \sum_{s,a\in\mathcal{C}(S,A^h)} \sum_{i=1}^{N^{K,H_h}(s,a)} r(\pi^*_{s,a}) - r(\pi^i_{s,a}) + \kappa \sum_{i=1}^{N^{K,H_h}(s,a)} P(s_{H_l}^{\pi^i_{s,a}} \neq g(s_h^k, a_h^k)) \quad (\kappa \geq 2H_h H_l)$$

$$= \sum_{s,a\in\mathcal{C}(S,A^h)} \sum_{i=1}^{N^{K,H_h}(s,a)} \frac{UB(\mathcal{R}^i(s,a)) - \mathcal{R}^i(s,a) + \kappa\psi_i}{i} + \sum_{s,a\in\mathcal{C}(S,A^h)} \mathcal{R}^{N^{K,H_h}(s,a)}$$
$$\text{(using the definition for sub-MDP regret)}$$

$$\leq \sum_{s,a\in\mathcal{C}(S,A^h)} \sum_{i=1}^{N^{K,H_h}(s,a)} \frac{UB(\mathcal{R}^i(s,a))}{i} + \mathcal{R}^{N^{K,H_h}(s,a)} + \sum_{s,a\in\mathcal{C}(S,A^h)} \sum_{i=1}^{N^{K,H_h}(s,a)} \frac{\kappa\psi_i}{i}$$

$$\leq \sum_{s,a\in\mathcal{C}(S,A^h)} \sum_{i=1}^{N^{K,H_h}(s,a)} \frac{UB(\mathcal{R}^i(s,a))}{i} + UB(\mathcal{R}^{N^{K,H_h}(s,a)}) + \sum_{s,a\in\mathcal{C}(S,A^h)} O(\kappa\sqrt{N^{K,H_h}(s,a)})$$
$$\text{(since Azuma-Hoeffding is s.t } \psi_i = O(\sqrt{i}))$$

$$\leq \sum_{s,a\in\mathcal{C}(S,A^h)} \sum_{i=1}^{N^{K,H_h}(s,a)} \frac{UB(\mathcal{R}^{N^{K,H_h}(s,a)})}{i} + UB(\mathcal{R}^{N^{K,H_h}(s,a)}) + \sum_{s,a\in\mathcal{C}(S,A^h)} O(H^h H^l \sqrt{N^{K,H_h}(s,a)})$$
$$\text{(using monotonicity of upper bound } UB(\mathcal{R}^i(s,a)) \text{ in } i, \text{ assumption that } C = O(H_h H_l))$$

$$= \sum_{s,a\in\mathcal{C}(S,A^h)} (\log(N^{K,H_h}(s,a))+1)UB(\mathcal{R}^{N^{K,H_h}(s,a)}) + O(H^h H^l \sqrt{N^{K,H_h}(s,a)})$$

$$\square$$

**Corollary 3** (Regret under $|\mathcal{C}(S, A^h)|$ clusters of isomorphic sub-MDPs [25]). *Let us set UCB-VI to be the sub-MDP learning algorithm, then we have the following regret bound:*

$$\sum_{s,a \in \mathcal{C}(S,A^h)} (\log(N^{K,H_h}(s,a)) + 1)\mathcal{R}^{N^{K,H_h}(s,a)} + O(H^h H^l \sqrt{N^{K,H_h}(s,a)})$$

$$\leq (\log H^h K + 1) \sum_{s,a \in \mathcal{C}(S,A^h)} \mathcal{R}^{N^{K,H_h}(s,a)} + O(H^h H^l \sqrt{|\mathcal{C}(S,A^h)| \cdot H^h K})$$

$$(\textstyle\sum_{s,a \in \mathcal{C}(S,A^h)} N^{K,H_h}(s,a) = H^h K)$$

$$\leq (\log H^h K + 1) \sum_{s,a \in \mathcal{C}(S,A^h)} H_l^{3/2} \sqrt{|S_{s,a}^l||A|N^{K,H_h}(s,a)} + O(H^h H^l \sqrt{|\mathcal{C}(S,A^h)| \cdot H^h K})$$

$$(\text{plug in UCB-VI guarantees})$$

$$\leq \tilde{O}(H_l^{3/2} \sqrt{\max_{s,a} |S_{s,a}^l||A|} \sqrt{|\mathcal{C}(S,A^h)|(H_h K)} + H_h H_l \sqrt{|\mathcal{C}(S,A^h)|H_h K})$$

$$(\textstyle\sum_{s,a \in \mathcal{C}(S,A^h)} N^{K,H_h}(s,a) = H^h K)$$

*using UCB-VI's guarantee that upper bound $UB(\mathcal{R}^{N^{K,H_h}(s,a)}) = H_l^{3/2} \sqrt{|S_{s,a}^l||A|N^{K,H_h}(s,a)}$.*

**Remark 3** (High Probability Bound). *For completeness, we show that the regret bound holds with probability greater than $1 - \delta$. The regret bound holds under $\bigcap_{s,a,n} \mathcal{E}_{s,a}^n \cap \mathcal{E}^\zeta \cap \mathcal{E}^\rho$, by union bound:*

$$\Pr\left(\bigcap_{s,a,n} \mathcal{E}_{s,a}^n \cap \mathcal{E}^\zeta \cap \mathcal{E}^\rho\right)$$

$$\geq 1 - \sum_{s,a,n} \Pr(\neg\mathcal{E}_{s,a}^n) - \Pr(\neg\mathcal{E}^\zeta) - \Pr(\neg\mathcal{E}^\rho))$$

$$\geq 1 - (|\mathcal{C}(S,A^h)|H_h K)\frac{\delta}{3|\mathcal{C}(S,A^h)|H_h K} - \delta/3 - \delta/3$$

$$= 1 - \delta$$

# D  Proofs for Section 4

## D.1  Low-level Feedback is insufficient for learning

To prove the results below, our approach is to construct two MDP instances with identical low level feedback such that any deterministic learning algorithm picks the arbitrarily worse high level policy.

**Proposition 4** (Non-identifiability of ranking among sub-MDP returns). *For any deterministic high-level policy learning algorithm with $N_l$ samples of low-level feedback, there exists a MDP instance that induces regret constant in $N_l$.*

*Proof.* Consider two-horizon MDP with starting state $s_1$ with $H_h = 1$, $H_l = 2$. There are two possible high-level actions $a_1$ and $a_2$ at $s_1$.

For any policy $\pi^1$ in sub-MDP $M(s_1, a_1)$, let it have feature expectation $\phi(\pi^1) = [\phi'(\pi^1), 1, 0]$, and for any $\pi^2$ in sub-MDP $M(s_1, a_2)$, $\phi(\pi^2) = [\phi'(\pi^2), 0, 1]$.

Now, we consider two MDP instances with $\theta^* = [0, 0, C']$ and $\theta^* = [0, C', 0]$ for some positive constant $C'$.

Under both instances, we observe identical low-level feedback for trajectories $\tau, \tau'$ in sub-MDPs $M(s_1, a_j)$, $j \in [2]$: the feedback is Bernoulli with parameter $\sigma(\langle \phi'(\tau) - \phi'(\tau), \theta' \rangle)$.

Consider any deterministic learning algorithm. WLOG it outputs high level policy $\pi^h(s_1) = a_1$ with some set of $N_l$ samples of low-level feedback.

Then, it follows that its regret under $\theta^* = [\epsilon 1, 0, C']$ is $C'$, since the reward (and return since $H_h = 1$) of $\pi^*_{s_1, a_1}$ is 0, while the reward of the optimal policy which visits $M(s_1, a_2)$ is $C'$.

$\square$

## D.2  Hierarchical Experiment Design via REGIME [29]

### D.2.1  MLE Definition:

We first define the MLE expression; note that the MLE is in terms of trajectories only. Define:

$$f(\{y_i\}_{i=1}^n, \{x_i\}_{i=1}^n) = -\sum_{i=1}^n \log(\mathbb{1}\{y_i = 1\}\sigma(\theta^T x_i) + \mathbb{1}\{y_i = 0\}(1 - \sigma(\theta^T x_i))$$

$$\ell_D(\theta) = f(\{y_i\}_{i=1}^{N_h}, \{x_i\}_{i=1}^n) + \sum_{s,a} f(\{y_i^{s,a}\}_{i=1}^{N_l}, \{x_i^{s,a}\}_{i=1}^{N_l}) \tag{5}$$

- **High-level trajectories:** has realized features,

$$x_i = \phi^{\pi^{N_l}, P}(\tau_1^i) - \phi^{\pi^{N_l}, P}(\tau_2^i) = \sum_{j=1}^{H_h} \phi^P(\pi^{N_l}(s_j^{\tau_1^i}, a_j^{\tau_1^i})) - \sum_{j=1}^{H_h} \phi^P(\pi^{N_l}(s_j^{\tau_2^i}, a_j^{\tau_2^i}))$$

  where $\phi^{\pi^{N_l}, P}(\tau_j^i)$ is the feature of the high-level trajectory under sub-policy $\pi^{N_l}$ and transition $P$ (since trajectories are collected from roll-outs in the actual MDP as in [29]).

  On the other hand, under idealized-feedback, the labeler assumes that each goal-conditioned sub-MDP has been executed perfectly (i.e. by $\pi^*_{s,a}$) and so the features correspond to:

$$x_i^* = \phi^{\pi^*, P}(\tau_1^i) - \phi^{\pi^*, P}(\tau_2^i) = \sum_{j=1}^{H_h} \phi^P(\pi^*(s_j^{\tau_1^i}, a_j^{\tau_1^i})) - \sum_{j=1}^{H_h} \phi^P(\pi^*(s_j^{\tau_2^i}, a_j^{\tau_2^i}))$$

- Comparison $y$ of high level trajectories follows Bernoulli distribution $y_i = \sigma(\theta^* \cdot x_i^*)$.

- **Low-level trajectories:** has realized features,

$$x_i^{s,a} = \phi(\tau_1^i) - \phi(\tau_2^i) = \sum_{j=1}^{H_h} \phi(s_j^{\tau_1^i}, a_j^{\tau_1^i}) - \sum_{j=1}^{H_h} \phi(s_j^{\tau_2^i}, a_j^{\tau_2^i})$$

Note that unlike the high level features, low-level features data are always unbiased. Thus, using high level and low-level comparisons has the same bias from the high level.

- Comparison $y$ of low level trajectories follows Bernoulli distribution $y_i = \sigma(\theta^* \cdot x_i^{s,a})$.

### D.2.2 Requisite Lemmas

**Lemma 10** (Lemma 5 of [29]). *Let oracle $P^{\epsilon'}$ be such that with probability $1 - \delta/5$, the following holds. Let $d_h^\pi(s, a)$ and $\hat{d}_h^\pi(s, a)$ be the visitation measure of policy $\pi$ under $P$ and $P^{\epsilon'}$, we have for all $h \in [H]$ and $\pi \in \Pi$:*

$$\sum_{s,a} |d_h^\pi(s, a) - \hat{d}_h^\pi(s, a)| = \sum_s |d_h^\pi(s) - \hat{d}_h^\pi(s)| \le h\epsilon'$$

This applies across all sub-MDPs $M(s, a)$. Let the event that this expression hold be $\mathcal{E}^{s,a}$.

**Lemma 11** (Low-level MLE Bound, Lemma 2 of [29]). *With probability at least $1 - \delta/5$:*

$$\|\theta^* - \theta^t\|_{\tilde{\Sigma}_n^l} \le \tilde{O}(1)$$

Let the event that this holds for learning from sub-MDP trajectories be $\mathcal{E}_1^l$.

**Lemma 12** (Lemma 3 of [29]). *If low-leve trajectories $\tau_i^{1,2} \sim \pi^i, P^{\epsilon'}$, then with probability at least $1 - \delta/5$:*

$$\|\theta^* - \theta^t\|_{\hat{\Sigma}_n^l} \le \sqrt{2}\|\theta^* - \theta^t\|_{\tilde{\Sigma}_n^l} + O(B\sqrt{d \log 4n/\delta}W)$$

Let the event that this holds for learning from sub-MDP trajectories be $\mathcal{E}_2^l$.

### D.2.3 Bias when using idealized-feedback, high level trajectory data in MLE

**Proposition 5** (sub-MDP REGIME guarantee of [29]). *For sub-MDP $M(s, a)$, under $\mathcal{E}^{s,a} \cap \mathcal{E}_1^l \cap \mathcal{E}_2^l$:*

$$\langle \phi^P(\pi^*), \theta^* \rangle - \langle \phi^P(\pi^{N_l}), \theta^* \rangle \le \frac{C_1(\delta)}{\sqrt{N_l}} + O(\epsilon')$$

*where $C_1(\delta) = O(\sqrt{\log(1/\delta)})$.*

Note that for estimation and bias, we have to have both an upper bound and a lower bound (see PbRL example). This requires two-sided bound, where lower bound comes from $\phi^*$ having higher reward than $\hat{\phi}$ and upper bound comes from no-regret. Due to optimality of $\pi^*$, we have the lower bound as well:

$$0 \le \langle \phi^P(\pi^*), \theta^* \rangle - \langle \phi^P(\pi^{N_l}), \theta^* \rangle \le \frac{C_1}{\sqrt{N_l}} + O(\epsilon')$$

Additionally, we have that:

**Lemma 13** (Lemma 6 of [29]). *For any $s_h, a_h$, $\|v_i\| \le 2B$, $\theta \in \mathbb{R}^d$ and $\|\phi\| \le R$ under $\mathcal{E}^{s,a} \cap \mathcal{E}_1^l \cap \mathcal{E}_2^l$:*

$$|\langle \phi^{P^{\epsilon'}}(\pi^{N_l}(s_h, a_h)) - \phi^P(\pi^{N_l}(s_h, a_h)), v \rangle| \le BRd^2\epsilon'$$

With this,

$$|\langle \phi^P(\pi^*), \theta^* \rangle - \phi^{P^{\epsilon'}}(\pi^{N_l}), \theta^*| \le (\frac{C_1}{\sqrt{N_l}} + O(\epsilon')) + BRd^2\epsilon' = \frac{C_1}{\sqrt{N_l}} + C_2\epsilon'$$

Now, we can analyze the bias of including high level trajectory data in the MLE computation:

**Lemma 14.** *Suppose there are $N_h, N_l$ high, low-level trajectories, bias $b$ is such that, under $\bigcap_{s,a} \mathcal{E}^{s,a} \cap \mathcal{E}_1^l \cap \mathcal{E}_2^l$:*

$$\|b\|^2 = \sum_{t=1}^T |\langle \theta^*, x_i \rangle - \langle \theta^*, x_i^* \rangle|^2 \le 2H_hT(2H_h(\frac{C_1}{\sqrt{N_l}} + C_2\epsilon')^2)$$

*Proof.*

$$\sum_{t=1}^{T} |\langle \theta^*, x_i^* \rangle - \langle \theta^*, x_i \rangle|^2$$

$$\leq 2 \sum_{t=1}^{T} |\langle \sum_{s,a \in \tau_1^t} \phi^P(\pi^*(s,a)) - \sum_{s,a \in \tau_1^t} \phi^{P^{\epsilon'}}(\pi^{N_l}(s,a)), \theta^* \rangle|^2 + |\langle \sum_{s,a \in \tau_2^t} \phi^P(\pi^*(s,a)) - \sum_{s,a \in \tau_2^t} \phi^{P^{\epsilon'}}(\pi^{N_l}(s,a)), \theta^* \rangle|^2$$

$$\leq 2 H_h \sum_{t=1}^{T} \sum_{s,a \in \tau_1^t} |\langle \phi^P(\pi^*(s,a)) - \phi^{P^{\epsilon'}}(\pi^{N_l}(s,a)), \theta^* \rangle|^2 + \sum_{s,a \in \tau_2^t} |\langle \phi^P(\pi^*(s,a)) - \phi^{P^{\epsilon'}}(\pi^{N_l}(s,a)), \theta^* \rangle|^2$$

$$\leq 2 H_h T (2 H_h (\frac{C_1}{\sqrt{N_l}} + C_2 \epsilon')^2)$$

Thus,

$$\|b\| = \sqrt{\sum_{t=1}^{T} |\langle \theta^*, x_i \rangle - \langle \theta^*, x_i^* \rangle|^2} \leq 2 H_h (\frac{C_1}{\sqrt{N_l}} + C_2 \epsilon') \sqrt{T}$$

$\square$

### D.2.4 MLE Analysis

Under current-feedback, following Lemma 2 of [29], $\|\Delta\|_{\Sigma_n^h + \lambda I} \leq \tilde{O}(1)$. Now, we consider the bias in learned reward under idealized-feedback.

**Proposition 6.** *Let $\theta_{MLE} = \arg\min_\theta \ell_D(\theta)$ and let $C_b \geq \|b\|$. Then with probability at least $1 - \delta/5$:*

$$\|\Delta\|_{\Sigma_n + \lambda I} \leq O\left(\sqrt{\frac{C_b}{\gamma^2 \sqrt{n}} + \frac{C_b^2 + d + \log(1/\delta)}{\gamma^2 n} + \lambda B^2}\right)$$

*where $\Sigma_n = \frac{1}{n} \sum_{i=1}^n x_i x_i^T + \lambda I$.*

*Proof.* Define $\Delta = \theta_{MLE} - \theta^*$. As in [30], we have the same convexity result due to $\langle \theta, x_i \rangle \in [-2LB, 2LB]$. Suppose we let $\max_x \|x\| \leq L$ and $\max_{\theta \in \Theta} \|\theta\| \leq B$, then with $\gamma = \frac{1}{2 + \exp(-2LB) + \exp(2LB)}$, we have that:

$$\ell(\theta^* + \Delta) - \ell(\theta^*) - \langle \nabla \ell(\theta^*), \Delta \rangle \geq \gamma \|\Delta\|_\Sigma^2$$

And so,

$$\ell(\theta_{MLE}) \leq \ell(\theta^*) \Rightarrow \ell(\theta^* + \Delta) - \ell(\theta^*) - \langle \nabla \ell(\theta^*), \Delta \rangle \leq -\langle \nabla \ell(\theta^*), \Delta \rangle$$

Thus,

$$\gamma \|\Delta\|_\Sigma^2 \leq \|\nabla \ell(\theta^*)\|_{(\Sigma + \lambda I)^{-1}} \|\Delta\|_{(\Sigma + \lambda I)}$$

The key part is bounding $\|\nabla \ell(\theta^*)\|_{(\Sigma + \lambda I)^{-1}}$. We have that:

$$\nabla \ell(\theta^*) = -\frac{1}{n} \sum_{i=1}^n [\mathbb{1}\{y_i = 1\} \sigma(\langle \theta^*, x_i \rangle) - \mathbb{1}\{y_i = 0\}(1 - \sigma(\langle \theta^*, x_i \rangle))] x_i$$

$$= -\frac{1}{n} X^T (V + b)$$

where $v_i = \sigma(\langle \theta^*, x_i^* \rangle)$ w.p $1 - \sigma(\langle \theta^*, x_i^* \rangle)$ and $-(1 - \sigma(\langle \theta^*, x_i^* \rangle))$ w.p $\sigma(\langle \theta^*, x_i^* \rangle)$. And so, entry-wise $V$ is such that $\mathbb{E}[V_i] = 0$ and $|V_i| \leq 1$. Note that $V_i$ are independent due to the independence of the random variables $Y_i$.

Extra term bias is defined as:

$$b_i = \mathbb{1}\{y_i = 1\}(\sigma(\langle \theta^*, x_i \rangle) - \sigma(\langle \theta^*, x_i^* \rangle)) - \mathbb{1}\{y_i = 0\}(1 - \sigma(\langle \theta^*, x_i \rangle) - (1 - \sigma(\langle \theta^*, x_i^* \rangle)))$$
$$= \sigma(\langle \theta^*, x_i \rangle) - \sigma(\langle \theta^*, x_i^* \rangle)$$

By definition, $C_b$ is such that: $\|b\| \leq C_b$. As before, define $M = \frac{1}{n^2} X (\Sigma + \lambda I)^{-1} X^T$. We use the fact that $\|M\|_{op} \leq 1/n$. Then, we have that:

$$\|\nabla\ell(\theta^*)\|^2_{(\Sigma+\lambda I)^{-1}} = (V+b)^T M(V+b)$$
$$= V^T M V + 2V^T M b + b^T M b$$
$$\leq C\frac{d+\log(1/\delta)}{n} + 2\|V\|\|Mb\| + b^T M b$$
$$\text{(by Matrix Bernstein, } V^T M V \leq C\frac{d+\log(10/\delta)}{n} \text{ w.p. } \geq 1-\delta/10)$$
$$\leq C\frac{d+\log(1/\delta)}{n} + 2\|V\|\frac{1}{n}\|b\| + \frac{C_b^2}{n} \qquad \text{(using that } \|M\|_{op} \leq 1/n)$$
$$\leq C\frac{d+\log(1/\delta)}{n} + 2(C_2\sqrt{n}\frac{1}{n})C_b + \frac{C_b^2}{n}$$
$$\text{(by Hoeffding } \|V\| \leq O(\log(10/\delta)\sqrt{n}) \text{ w.p. } \geq 1-\delta/10.)$$
$$\leq O(\frac{C_b}{\sqrt{n}} + \frac{C_b^2+d+\log(1/\delta)}{n})$$

$$\gamma\|\Delta\|^2_{\Sigma+\lambda I} \leq \|\nabla\ell(\theta^*)\|_{(\Sigma+\lambda I)^{-1}}\|\Delta\|_{(\Sigma+\lambda I)} + \lambda(\gamma\|\Delta\|^2)$$
$$\leq \|\nabla\ell(\theta^*)\|_{(\Sigma+\lambda I)^{-1}}\|\Delta\|_{(\Sigma+\lambda I)} + 4\lambda\gamma B^2$$

This implies that with probability $\geq 1-\delta$:

$$\|\Delta\|_{\Sigma+\lambda I} \leq C\sqrt{\frac{C_b}{\gamma^2\sqrt{n}} + \frac{C_b^2+d+\log(1/\delta)}{\gamma^2 n} + \lambda B^2}$$

$\square$

**Corollary 4.** *Let* $\theta_{MLE} = \arg\min_\theta \ell_D(\theta)$, *then under* $\bigcap_{s,a} \mathcal{E}^{s,a}$, *with probability* $\geq 1-\delta/5$:

$$\|\theta^* - \theta_{MLE}\|_{\tilde{\Sigma}^h_{N^h}+\lambda I} \leq C\sqrt{\frac{1}{\gamma^2\sqrt{N_l}} + \frac{1}{\gamma^2 N_l} + \frac{d+\log(1/\delta)}{\gamma^2 N_h} + \lambda B^2}$$

*where* $\Sigma_{N_h} = \frac{1}{N_h}\sum_{i=1}^{N_h} x_i x_i^T$.

*Let the event that this holds for learning from sub-MDP trajectories be* $\mathcal{E}_1^h$.

*Proof.* Firstly,

$$\|b\| \leq 2H_h(\frac{C_1}{\sqrt{N_l}} + C_2\epsilon')\sqrt{N_h} = O(\frac{\sqrt{N_h}}{\sqrt{N_l}} + \sqrt{N_h}\epsilon')$$

With this, we have that:

$$\|\Delta\|_{\tilde{\Sigma}_{N_h}+\lambda I}$$
$$= O\left(\sqrt{\frac{C_b}{\gamma^2\sqrt{N_h}} + \frac{C_b^2+d+\log(1/\delta)}{\gamma^2 N_h} + \lambda B^2)}\right)$$
$$= O\left(\sqrt{\frac{\sqrt{N_h/N_l}+\sqrt{N_h}\epsilon'}{\gamma^2\sqrt{N_h}} + \frac{N_h/N_l+N_h\epsilon'^2+d+\log(1/\delta)}{\gamma^2 N_h} + \lambda B^2}\right)$$

$\square$

Hence by choosing $\lambda = \lambda/N_h$:

$$\|\Delta\|_{\tilde{\Sigma}_{N_h}+\lambda I} \leq O\left(\frac{N_h^{1/2}}{N_l^{1/4}} + (N_h\epsilon')^{1/2}\right) + C'$$

**D.2.5   Relating $\|\theta^* - \theta^n\|_{\hat{\Sigma}_n}$ to $\|\theta^* - \theta^n\|_{\tilde{\Sigma}_n}$**

Define:

1. $\Sigma_n = \lambda I + \sum_{i=1}^n (\phi^{\pi^{N_l},P}(\pi_1^i) - \phi^{\pi^{N_l},P}(\pi_2^i))(\phi^{\pi^{N_l},P}(\pi_1^i) - \phi^{\pi^{N_l},P}(\pi_2^i))^T$
2. $\tilde{\Sigma}_n = \lambda I + \sum_{i=1}^n (\phi(\tau_1^i) - \phi(\tau_2^i))(\phi(\tau_1^i) - \phi(\tau_2^i))^T$, where $\tau_i^{1,2} \sim \pi_1^i, \pi^{N_l}, P$.
3. $\hat{\Sigma}_n = \lambda I + \sum_{i=1}^n (\phi^{\pi^{N_l},P^{\epsilon'}}(\pi_1^i) - \phi^{\pi^{N_l},P^{\epsilon'}}(\pi_2^i))(\phi^{\pi^{N_l},P^{\epsilon'}}(\pi_1^i) - \phi^{\pi^{N_l},P^{\epsilon'}}(\pi_2^i))^T$

We wish to relate $\|\theta^* - \theta^n\|_{\hat{\Sigma}_n}$ to $\|\theta^* - \theta^n\|_{\tilde{\Sigma}_n}$.

**Lemma 15** (Lemma 3 of [29]). *If $\tau_i^{1,2} \sim \pi_1^i, \pi^{N_l}, P^{\epsilon'}$, then with probability at least $1 - \delta/5$:*

$$\|\theta^* - \theta^t\|_{\hat{\Sigma}_n^h} \leq \sqrt{2}\|\theta^* - \theta^t\|_{\tilde{\Sigma}_n^h} + \tilde{O}(B\sqrt{d\log 4n/\delta}W)$$

*Let the event that this holds for learning from sub-MDP trajectories be $\mathcal{E}_2^h$.*

**Lemma 16.** *We have that under $\bigcap_{s,a} \mathcal{E}^{s,a} \cap \mathcal{E}_1^l \cap \mathcal{E}_2^l \cap \mathcal{E}_1^h \cap \mathcal{E}_2^h$:*

$$\|\theta^* - \theta^n\|_{\hat{\Sigma}_n} \leq 2\|\theta^* - \theta^n\|_{\tilde{\Sigma}_n} + O(B\sqrt{d\log n/\delta}W) + \sqrt{8n}C(\epsilon',\delta)$$

*Proof.* Under event $\mathcal{E}_2^h$, as trajectories are sampled from $P$, we have that:

$$\|\theta^* - \theta^n\|_{\Sigma_n} \leq \sqrt{2}\|\theta^* - \theta^n\|_{\tilde{\Sigma}_n} + O(B\sqrt{d\log n/\delta}W)$$

It remains to upper bound $\|\theta^* - \theta^n\|_{\hat{\Sigma}_n}$ by $\|\theta^* - \theta^n\|_{\Sigma_n}$

We have that under $\bigcap_{s,a} \mathcal{E}^{s,a} \cap \mathcal{E}_1^l \cap \mathcal{E}_2^l$:

$$|\langle \phi^{\pi^{N_l},P}(\pi) - \phi^{\pi^{N_l},P^{\epsilon'}}(\pi), v\rangle| \leq C(\epsilon',\delta)$$
$$\Rightarrow |\langle \phi^{\pi^{N_l},P^{\epsilon'}}(\pi_1^i) - \phi^{\pi^{N_l},P^{\epsilon'}}(\pi_2^i), v\rangle| \leq |\langle \phi^{\pi^{N_l},P}(\pi_1^i) - \phi^{\pi^{N_l},P}(\pi_2^i), v\rangle| + 2C(\epsilon',\delta)$$
$$\Rightarrow |\langle \phi^{\pi^{N_l},P^{\epsilon'}}(\pi_1^i) - \phi^{\pi^{N_l},P^{\epsilon'}}(\pi_2^i), v\rangle|^2 \leq 2|\langle \phi^{\pi^{N_l},P}(\pi_1^i) - \phi^{\pi^{N_l},P}(\pi_2^i), v\rangle|^2 + 2(2C(\epsilon',\delta))^2$$

Thus,

$$\begin{aligned}
&\|v\|_{\hat{\Sigma}_n}^2\\
&= v^T(\lambda I + \sum_{i=1}^n (\phi^{\pi^{N_l},P^{\epsilon'}}(\pi_1^i) - \phi^{\pi^{N_l},P^{\epsilon'}}(\pi_2^i))(\phi^{\pi^{N_l},P^{\epsilon'}}(\pi_1^i) - \phi^{\pi^{N_l},P^{\epsilon'}}(\pi_2^i))^T)v\\
&= \lambda\|v\|^2 + \sum_{i=1}^n |\langle \phi^{\pi^{N_l},P^{\epsilon'}}(\pi_1^i) - \phi^{\pi^{N_l},P^{\epsilon'}}(\pi_2^i), v\rangle|^2\\
&\leq \lambda\|v\|^2 + \sum_{i=1}^n 2|\langle \phi^{\pi^{N_l},P}(\pi_1^i) - \phi^{\pi^{N_l},P}(\pi_2^i), v\rangle|^2 + 8C(\epsilon',\delta)^2\\
&\leq 2\|v\|_{\Sigma_n}^2 + 8nC(\epsilon',\delta)^2
\end{aligned}$$

Plugging in $v = \theta^* - \theta^n$, we have that:

$$\|\theta^* - \theta^n\|_{\hat{\Sigma}_n}$$

$$\leq \sqrt{2}\|\theta^* - \theta^n\|_{\Sigma_n} + \sqrt{8n}C(\epsilon', \delta)$$

$$\leq 2\|\theta^* - \theta^n\|_{\tilde{\Sigma}_n} + O(B\sqrt{d\log n/\delta}W) + \sqrt{8n}C(\epsilon', \delta)$$

$\square$

### D.2.6 High-level policy regret bound

**Lemma 17.** *For any $\pi$, under event $\bigcap_{s,a} \mathcal{E}^{s,a} \cap \mathcal{E}_1^l \cap \mathcal{E}_2^l$:*

$$\langle \phi^{\pi^*,P}(\pi) - \phi^{\pi^{N_l},P}(\pi), \theta^* \rangle \leq H_h(\frac{C_1}{\sqrt{N_l}} + C_2\epsilon')$$

*Proof.*

$\langle \phi^{\pi^*,P}(\pi) - \phi^{\pi^{N_l},P}(\pi), \theta^* \rangle$

$$= \sum_{h=1}^{H_h} \mathbb{E}_{s_h,a_h \sim \pi,\pi^{N_l},P} \mathbb{E}_{s_{h+1} \sim \pi^{N_l}(s_h,a_h),P}[r(\pi^*(s_h,a_h)) + V_{h+1}^{\pi,\pi^*}(g(s_h,a_h)) - (r(\pi^{N_l}(s_h,a_h)) + V_{h+1}^{\pi,\pi^{N_l}}(s_{h+1}))]$$

$$= \sum_{h=1}^{H_h} \mathbb{E}_{s_h,a_h \sim \pi,\pi^{N_l},P}[r(\pi^*(s_h,a_h)) - r(\pi^{N_l}(s_h,a_h)) + P(s_{h+1}^{\pi^{N_l}} \neq g(s_h,a_h))(V_{h+1}^{\pi,\pi^*}(g(s_h,a_h)) - V_{h+1}^{\pi,\pi^{N_l}}(s_{h+1}))]$$

$$\leq \sum_{h=1}^{H_h} \mathbb{E}_{s_h,a_h \sim \pi,\pi^{N_l},P}[r(\pi^*(s_h,a_h)) - r(\pi^{N_l}(s_h,a_h)) + P(s_{h+1}^{\pi^{N_l}} \neq g(s_h,a_h))\kappa H_h H_l]$$

$$= \sum_{h=1}^{H_h} \mathbb{E}_{s_h,a_h \sim \pi,\pi^{N_l},P}[r(\pi^*(s_h,a_h)) + P(s_{h+1}^{\pi^*} = g(s_h,a_h))\kappa H_h H_l - r(\pi^{N_l}(s_h,a_h)) - P(s_{h+1}^{\pi^{N_l}} = g(s_h,a_h))\kappa H_h H_l]$$

$$= \sum_{h=1}^{H_h} \mathbb{E}_{s_h,a_h \sim \pi,\pi^{N_l},P}[\langle \phi(\pi^*(s_h,a_h)), \theta^* \rangle - \langle \phi(\pi^{N_l}(s_h,a_h)), \theta^* \rangle]$$

$$\leq H_h(\frac{C_1}{\sqrt{N_l}} + C_2\epsilon')$$

Because for any $s_h, a_h$, $\langle \phi(\pi^*(s_h,a_h)), \theta^* \rangle - \langle \phi(\pi^{N_l}(s_h,a_h)), \theta^* \rangle \leq \frac{C_1}{\sqrt{N_l}} + C_2\epsilon'$.

$\square$

**Lemma 18** (Lower bound on Reachability Probability). *We have that under event $\bigcap_{s,a} \mathcal{E}^{s,a} \cap \mathcal{E}_1^l \cap \mathcal{E}_2^l$:*

$$P(s_{H_l}^{\pi^{N_l}} \neq g(s,a)) \leq \frac{1}{\kappa H_h} + \frac{C_1}{\kappa H_h H_l \sqrt{N_l}} + \frac{C_2 \epsilon'}{\kappa H_h H_l}$$

*and*

$$P^{\epsilon'}(s_{H_l}^{\pi^{N_l}} \neq g(s,a)) \leq \frac{1}{\kappa H_h} + \frac{C_1}{\kappa H_h H_l \sqrt{N_l}} + \frac{C_2 \epsilon'}{\kappa H_h H_l} + H_l \epsilon'$$

*Proof.* Due to the regret guarantee, we have that:

$$
\begin{aligned}
& \frac{C_1}{\sqrt{N_l}} + C_2 \epsilon' \\
& \geq \langle \phi^P(\pi^*) - \phi^P(\pi^{N_l}), \theta^* \rangle \\
& = r(\pi^*) + \kappa H_h H_l \cdot 1 - r(\pi^{N_l}) - \kappa H_h H_l \cdot P(s_{H_l}^{\pi^{N_l}} = g(s,a)) \\
& \geq 0 - H_l + \kappa H_h H_l \cdot P(s_{H_l}^{\pi^{N_l}} \neq g(s,a))
\end{aligned}
$$

Thus, we have that:

$$P(s_{H_l}^{\pi^{N_l}} \neq g(s,a)) \leq \frac{1}{\kappa H_h} + \frac{C_1}{\kappa H_h H_l \sqrt{N_l}} + \frac{C_2 \epsilon'}{\kappa H_h H_l}$$

Additionally, we have that from Lemma 5.1:

$$|d_{H_l}^{\pi^{N_l}}(g(s,a)) - \hat{d}_{H_l}^{\pi^{N_l}}(g(s,a))| = |P(s_{H_l}^{\pi^{N_l}} \neq g(s,a)) - P^{\epsilon'}(s_{H_l}^{\pi^{N_l}} \neq g(s,a))| \leq H_l \epsilon'$$

Thus,

$$P^{\epsilon'}(s_{H_l}^{\pi^{N_l}} \neq g(s,a)) \leq \frac{1}{\kappa H_h} + \frac{C_1}{\kappa H_h H_l \sqrt{N_l}} + \frac{C_2 \epsilon'}{\kappa H_h H_l} + H_l \epsilon'$$

$\square$

Define goal non-reachability probability to be: $\delta = \frac{1}{\kappa H_h} + \frac{C_1}{\kappa H_h H_l \sqrt{N_l}} + \frac{C_2 \epsilon'}{\kappa H_h H_l} + H_l \epsilon'$.

**Lemma 19.** *Let $\Phi^{\pi^{N_l}, P^{\epsilon'}}(\pi)$ denote the feature expectation under high level policy $\pi$, sub-MDP policies $\pi^{N_l}$ and MDP transitions $P^{\epsilon'}$. Under event $\bigcap_{s,a} \mathcal{E}^{s,a} \cap \mathcal{E}_1^l \cap \mathcal{E}_2^l$, we have that, for any high level policy $\pi$:*

$$|\langle \phi^{\pi^{N_l}, P}(\pi) - \phi^{\pi^{N_l}, P^{\epsilon'}}(\pi), \theta^* \rangle| \leq 2 H_h B R d^2 \epsilon' + 8 H_h^3 H_l \delta$$

*Proof.* Let $\mathcal{E}_{reach}$ denote the event that roll-out $\tau \sim \pi, \pi^{N_l}, P$ is such that all high level goals are reached, and similarly event $\mathcal{E}'_{reach}$ for roll-out $\tau' \sim \pi, \pi^{N_l}, P^{\epsilon'}$.

By union bound, $\Pr(\neg \mathcal{E}_{reach}) = \Pr(\exists s_i, a_i, s_{H_l}^{\pi^{N_l}(s_i, a_i)} \neq g(s_i, a_i)) \leq \sum_{i=1}^{H_h} \Pr(s_{H_l}^{\pi^{N_l}(s_i, a_i)} \neq g(s_i, a_i)))) \leq H_h \delta$, and similarly $\Pr(\neg \mathcal{E}'_{reach}) \leq H_h \delta$.

$$|\langle \phi^{\pi^{N_l},P}(\pi) - \phi^{\pi^{N_l},P^{\epsilon'}}(\pi), \theta^*\rangle|$$

$$\leq |\mathbb{E}_{\tau\sim\pi,\pi^{N_l},P}[\langle\phi(\tau),\theta^*\rangle|\mathcal{E}_{reach}]\Pr(\mathcal{E}_{reach}) - \mathbb{E}_{\tau\sim\pi,\pi^{N_l},P^{\epsilon'}}[\langle\phi(\tau),\theta^*\rangle|\mathcal{E}'_{reach}]\Pr(\mathcal{E}'_{reach})|$$

$$+ |\mathbb{E}_{\tau\sim\pi,\pi^{N_l},P}[\langle\phi(\tau),\theta^*\rangle|\neg\mathcal{E}_{reach}]\Pr(\neg\mathcal{E}_{reach}) - \mathbb{E}_{\tau\sim\pi,\pi^{N_l},P^{\epsilon'}}[\langle\phi(\tau),\theta^*\rangle|\neg\mathcal{E}'_{reach}]\Pr(\neg\mathcal{E}'_{reach})|$$

$$\leq |\mathbb{E}_{\tau\sim\pi,\pi^{N_l},P}[\langle\phi(\tau),\theta^*\rangle|\mathcal{E}_{reach}]\Pr(\mathcal{E}_{reach}) - \mathbb{E}_{\tau\sim\pi,\pi^{N_l},P^{\epsilon'}}[\langle\phi(\tau),\theta^*\rangle|\mathcal{E}'_{reach}]\Pr(\mathcal{E}'_{reach})| + 2(H_h\delta)(H_hH_l)$$

(since $|\mathbb{E}_{\tau\sim\pi,\pi^{N_l},P}[\langle\phi(\tau),\theta^*\rangle|\neg\mathcal{E}_{reach}]\Pr(\neg\mathcal{E}_{reach})| \leq (H_h\delta)(H_hH_l)$ and likewise the other term)

$$= |\Pr(\mathcal{E}_{reach})\sum_{h=1}^{H_h}\sum_{s_h,a_h}d(s_h,a_h)\mathbb{E}[\langle\phi^P(\pi^{N_l}(s_h,a_h)),\theta^*\rangle|\mathcal{E}_{reach}]$$

$$- \Pr(\mathcal{E}'_{reach})\sum_{h=1}^{H_h}\sum_{s_h,a_h}d(s_h,a_h)\mathbb{E}[\langle\phi^{P^{\epsilon'}}(\pi^{N_l}(s_h,a_h)),\theta^*\rangle|\mathcal{E}'_{reach}]| + 2H_h^2H_l\delta$$

(under goal reachability, high-level state visitation measure $d(s_h,a_h)$ is the same)

$$\leq \sum_{h=1}^{H_h}\sum_{s_h,a_h}d(s_h,a_h)|\Pr(\mathcal{E}_{reach})\mathbb{E}[\langle\phi^P(\pi^{N_l}(s_h,a_h)),\theta^*\rangle|\mathcal{E}_{reach}]$$

$$- \Pr(\mathcal{E}'_{reach})\mathbb{E}[\langle\phi^{P^{\epsilon'}}(\pi^{N_l}(s_h,a_h)),\theta^*\rangle|\mathcal{E}'_{reach}]| + 2H_h^2H_l\delta$$

$$= \sum_{h=1}^{H_h}\sum_{s_h,a_h}d(s_h,a_h)|\Pr(\mathcal{E}_{reach})\mathbb{E}[\langle\phi^P(\pi^{N_l}(s_h,a_h)),\theta^*\rangle|\mathcal{E}_{s_h,a_h reach}]$$

$$- \Pr(\mathcal{E}'_{reach})\mathbb{E}[\langle\phi^{P^{\epsilon'}}(\pi^{N_l}(s_h,a_h)),\theta^*\rangle|\mathcal{E}'_{s_h,a_h reach}]| + 2H_h^2H_l\delta$$

($\mathcal{E}_{s_h,a_h reach}$ is the event that $g(s_h,a_h)$ is reached under $\pi^{N_l},P$)

$$\leq \sum_{h=1}^{H_h}\sum_{s_h,a_h}d(s_h,a_h)\Pr(\mathcal{E}_{reach})|\mathbb{E}[\langle\phi^P(\pi^{N_l}(s_h,a_h)),\theta^*\rangle|\mathcal{E}_{s_h,a_h reach}] - \mathbb{E}[\langle\phi^{P^{\epsilon'}}(\pi^{N_l}(s_h,a_h)),\theta^*\rangle|\mathcal{E}'_{s_h,a_h reach}]|$$

$$+ |(\Pr(\mathcal{E}_{reach}) - \Pr(\mathcal{E}'_{reach}))\mathbb{E}[\langle\phi^{P^{\epsilon'}}(\pi^{N_l}(s_h,a_h)),\theta^*\rangle|\mathcal{E}'_{s_h,a_h reach}]| + 2H_h^2H_l\delta$$

$$\leq \sum_{h=1}^{H_h}\sum_{s_h,a_h}d(s_h,a_h)\left(|\mathbb{E}[\langle\phi^P(\pi^{N_l}(s_h,a_h)),\theta^*\rangle|\mathcal{E}_{s_h,a_h reach}] - \mathbb{E}[\langle\phi^{P^{\epsilon'}}(\pi^{N_l}(s_h,a_h)),\theta^*\rangle|\mathcal{E}'_{s_h,a_h reach}]| + (H_h\delta)(H_hH_l)\right)$$

$$+ 2H_h^2H_l\delta \qquad\qquad\qquad\qquad \text{(since } \Pr(\mathcal{E}'_{reach}), \Pr(\mathcal{E}_{reach}) \in [1 - H_h\delta, 1])$$

To finish, we will relate the expression to $|\langle\phi^{P^{\epsilon'}}(\pi^{N_l}(s_h,a_h)) - \phi(\pi^{N_l}(s_h,a_h)),\theta^*\rangle|$.

$$\leq \sum_{h=1}^{H_h} \sum_{s_h,a_h} d(s_h,a_h)|\mathbb{E}[\langle \phi^P(\pi^{N_l}(s_h,a_h)),\theta^*\rangle|\mathcal{E}_{s_h,a_h reach}] - \mathbb{E}[\langle \phi^{P^{\epsilon'}}(\pi^{N_l}(s_h,a_h)),\theta^*\rangle|\mathcal{E}'_{s_h,a_h reach}]| + 3H_h^3 H_l \delta$$

$$= \sum_{h=1}^{H_h} \sum_{s_h,a_h} d(s_h,a_h)|\frac{1}{\Pr(\mathcal{E}_{s_h,a_h reach})}\Pr(\mathcal{E}_{s_h,a_h reach})\mathbb{E}[\langle \phi^P(\pi^{N_l}(s_h,a_h)),\theta^*\rangle|\mathcal{E}_{s_h,a_h reach}]$$

$$- \frac{1}{\Pr(\mathcal{E}'_{s_h,a_h reach})}\Pr(\mathcal{E}'_{s_h,a_h reach})\mathbb{E}[\langle \phi^{P^{\epsilon'}}(\pi^{N_l}(s_h,a_h)),\theta^*\rangle|\mathcal{E}'_{s_h,a_h reach}]| + 3H_h^3 H_l \delta$$

$$\leq \sum_{h=1}^{H_h} \sum_{s_h,a_h} d(s_h,a_h)\frac{1}{\Pr(\mathcal{E}_{s_h,a_h reach})}|\Pr(\mathcal{E}_{s_h,a_h reach})\mathbb{E}[\langle \phi^P(\pi^{N_l}(s_h,a_h)),\theta^*\rangle|\mathcal{E}_{s_h,a_h reach}]$$

$$- \Pr(\mathcal{E}'_{s_h,a_h reach})\mathbb{E}[\langle \phi^{P^{\epsilon'}}(\pi^{N_l}(s_h,a_h)),\theta^*\rangle|\mathcal{E}'_{s_h,a_h reach}]| + H_h\left((\frac{1}{1-\delta}-1)H_h H_l\right) + 3H_h^3 H_l \delta$$

$$(\diamond)$$

$$\leq \sum_{h=1}^{H_h} \sum_{s_h,a_h} d(s_h,a_h)\frac{1}{1-\delta}|\Pr(\neg\mathcal{E}_{s_h,a_h reach})\mathbb{E}[\langle \phi^P(\pi^{N_l}(s_h,a_h)),\theta^*\rangle|\neg\mathcal{E}_{s_h,a_h reach}]$$

$$- \Pr(\neg\mathcal{E}'_{s_h,a_h reach})\mathbb{E}[\langle \phi^{P^{\epsilon'}}(\pi^{N_l}(s_h,a_h)),\theta^*\rangle|\neg\mathcal{E}'_{s_h,a_h reach}]|+$$

$$|\mathbb{E}[\langle \phi^{P^{\epsilon'}}(\pi^{N_l}(s_h,a_h)) - \phi^P(\pi^{N_l}(s_h,a_h)),\theta^*\rangle]| + 4H_h^3 H_l \delta \qquad \text{(using that } \frac{1}{1-\delta}-1 \leq 1)$$

$$\leq \sum_{h=1}^{H_h} \sum_{s_h,a_h} d(s_h,a_h)\frac{1}{1-\delta}\left(2(\delta)(H_h H_l) + BRd^2\epsilon'\right) + 4H_h^3 H_l \delta \qquad (\diamond\diamond)$$

$$\leq \sum_{h=1}^{H_h} \sum_{s_h,a_h} d(s_h,a_h)2\left(2H_h H_l \delta + BRd^2\epsilon'\right) + 4H_h^3 H_l \delta \qquad (\frac{1}{1-\delta} \leq 2)$$

$$\leq 2H_h BRd^2\epsilon' + 8H_h^3 H_l \delta = C(\epsilon',\delta)$$

$(\diamond)$ : $|\frac{\Pr(\mathcal{E}'_{s_h,a_h reach})}{\Pr(\mathcal{E}_{s_h,a_h reach})} - 1| \leq \max(1-(1-\delta)\frac{1}{1-\delta}-1)$ since $\Pr(\mathcal{E}'_{s_h,a_h reach}), \Pr(\mathcal{E}_{s_h,a_h reach}) \in [1-\delta,1]$.

$(\diamond\diamond)$ : $|\langle \phi^{P^{\epsilon'}}(\pi^{N_l}(s_h,a_h)) - \phi^P(\pi^{N_l}(s_h,a_h)),v\rangle| \leq BRd^2\epsilon'$ and $\Pr(\neg\mathcal{E}_{s_h,a_h reach}), \Pr(\neg\mathcal{E}'_{s_h,a_h reach}) \in [0,\delta]$

$\square$

**Lemma 20** (use of the Elliptical Lemma).

$$\langle \phi^{\pi^{N_l},P^{\epsilon'}}(\pi^*) - \phi^{\pi^{N_l},P^{\epsilon'}}(\hat{\pi}), \theta^* - \hat{\theta}\rangle \leq \frac{1}{\sqrt{N_h}}(2d\log(1+\frac{N_h}{d}))\|\theta^* - \hat{\theta}\|_{\hat{\Sigma}_{N_h}}$$

*Proof.*

$$\langle \phi^{\pi^{N_l},P^{\epsilon'}}(\pi^*) - \phi^{\pi^{N_l},P^{\epsilon'}}(\hat{\pi}), \theta^* - \hat{\theta}\rangle$$

$$\leq \|\phi^{\pi^{N_l},P^{\epsilon'}}(\pi^*) - \phi^{\pi^{N_l},P^{\epsilon'}}(\hat{\pi})\|_{\hat{\Sigma}_{N_h}^{-1}}\|\theta^* - \hat{\theta}\|_{\hat{\Sigma}_{N_h}}$$

$$\leq \frac{1}{N_h}\sum_{i=1}^{N_h}\|\phi^{\pi^{N_l},P^{\epsilon'}}(\pi^*) - \phi^{\pi^{N_l},P^{\epsilon'}}(\hat{\pi})\|_{\hat{\Sigma}_i^{-1}}\|\theta^* - \hat{\theta}\|_{\hat{\Sigma}_{N_h}} \qquad (\hat{\Sigma}_{N_h}^{-1} \preceq \hat{\Sigma}_i^{-1})$$

$$\leq \frac{1}{N_h}\sum_{i=1}^{N_h}\|\phi^{\pi^{N_l},P^{\epsilon'}}(\pi_1^i) - \phi^{\pi^{N_l},P^{\epsilon'}}(\pi_2^i)\|_{\hat{\Sigma}_i^{-1}}\|\theta^* - \hat{\theta}\|_{\hat{\Sigma}_{N_h}} \qquad (\text{definition of } \pi_{1,2}^i)$$

$$\leq \frac{1}{\sqrt{N_h}}\sqrt{\sum_{i=1}^{N_h}\|\phi^{\pi^{N_l},P^{\epsilon'}}(\pi_1^i) - \phi^{\pi^{N_l},P^{\epsilon'}}(\pi_2^i)\|_{\hat{\Sigma}_i^{-1}}^2}\|\theta^* - \hat{\theta}\|_{\hat{\Sigma}_{N_h}}$$

$$\leq \frac{1}{\sqrt{N_h}}(2d\log(1+\frac{N_h}{d}))\|\theta^* - \hat{\theta}\|_{\hat{\Sigma}_{N_h}} \qquad (\text{Elliptical Lemma})$$

$\square$

**Theorem 4** (Main regret bound). *We have that under event $\bigcap_{s,a}\mathcal{E}^{s,a} \cap \mathcal{E}_1^l \cap \mathcal{E}_2^l \cap \mathcal{E}_1^h \cap \mathcal{E}_2^h$ and $N_h > 0$:*

$$V^{\pi^*,\pi^*} - V^{\hat{\pi},\pi^{N_l}} \leq \tilde{O}\left(N_l^{-1/2} + N_h^{-1/2}\|\theta^* - \hat{\theta}\|_{\hat{\Sigma}_{N_h}}\right)$$

*Proof.*

$$V^{\pi^*,\pi^*} - V^{\hat{\pi},\pi^{N_l}}$$

$$= \langle \phi^{\pi^*,P}(\pi^*) - \phi^{\pi^{N_l},P}(\hat{\pi}), \theta^*\rangle$$

$$= \langle \phi^{\pi^*,P}(\pi^*) - \phi^{\pi^{N_l},P}(\pi^*), \theta^*\rangle + \langle \phi^{\pi^{N_l},P}(\pi^*) - \phi^{\pi^{N_l},P}(\hat{\pi}), \theta^*\rangle$$
$$\qquad (\text{first term = sub-MDP sub-optimality; second term = high-level policy sub-optimality})$$

$$\leq H_h(\frac{C_1}{\sqrt{N_l}} + C_2\epsilon') + \langle \phi^{\pi^{N_l},P}(\pi^*) - \phi^{\pi^{N_l},P}(\hat{\pi}), \theta^*\rangle$$

$$\leq H_h(\frac{C_1}{\sqrt{N_l}} + C_2\epsilon') + \langle \phi^{\pi^{N_l},P^{\epsilon'}}(\pi^*) - \phi^{\pi^{N_l},P^{\epsilon'}}(\hat{\pi}), \theta^*\rangle$$

$$+ |\langle \phi^{\pi^{N_l},P}(\pi^*) - \phi^{\pi^{N_l},P^{\epsilon'}}(\pi^*), \theta^*\rangle| + |\langle \phi^{\pi^{N_l},P^{\epsilon'}}(\hat{\pi}) - \phi^{\pi^{N_l},P}(\hat{\pi}), \theta^*\rangle|$$

$$\leq H_h(\frac{C_1}{\sqrt{N_l}} + C_2\epsilon') + 2C(\epsilon',\delta) + \langle \phi^{\pi^{N_l},P^{\epsilon'}}(\pi^*) - \phi^{\pi^{N_l},P^{\epsilon'}}(\hat{\pi}), \theta^* - \hat{\theta}\rangle + \langle \phi^{\pi^{N_l},P^{\epsilon'}}(\pi^*) - \phi^{\pi^{N_l},P^{\epsilon'}}(\hat{\pi}), \hat{\theta}\rangle$$
$$\qquad (\text{expand out the second term})$$

$$\leq H_h(\frac{C_1}{\sqrt{N_l}} + C_2\epsilon') + 2C(\epsilon',\delta) + \langle \phi^{\pi^{N_l},P^{\epsilon'}}(\pi^*) - \phi^{\pi^{N_l},P^{\epsilon'}}(\hat{\pi}), \theta^* - \hat{\theta}\rangle$$
$$\qquad (\text{definition of } \hat{\pi}: \langle \phi^{\pi^{N_l},P^{\epsilon'}}(\pi^*) - \phi^{\pi^{N_l},P^{\epsilon'}}(\hat{\pi}), \hat{\theta}\rangle \leq 0)$$

$$\leq H_h(\frac{C_1}{\sqrt{N_l}} + C_2\epsilon') + 2C(\epsilon',\delta) + \frac{1}{\sqrt{N_h}}(2d\log(1+\frac{N_h}{d}))\|\theta^* - \hat{\theta}\|_{\hat{\Sigma}_{N_h}}$$
$$\qquad (\text{use of Elliptical lemma})$$

$\square$

**Data Tradeoff:** Using the above bound, we can derive the following rates:

- Under idealized-feedback and requiring both high- and low-level feedback, the overall rate comes out to $O(N_l^{-1/4} + N_h^{-1/2})$.

  This is because $\hat{\Sigma}_{N_h} = O\left(\frac{N_h^{1/2}}{N_l^{1/4}} + 1\right)$. Thus, the dominating factor is the bias of the reward learning.

- Under current-feedback and requiring both high- and low-level feedback, the overall rate comes out to $O(N_l^{-1/2} + N_h^{-1/2})$.

  This is because $\|\theta^* - \hat{\theta}\|_{\hat{\Sigma}_{N_h}} = O(1)$.

- Under only low-level feedback (due to sufficiency in coverage), the overall rate comes out to $O(N_l^{-1/2})$.

  We have that:

$$
\begin{aligned}
&\langle \phi^{\pi^{N_l}, P^{\epsilon'}}(\pi^*) - \phi^{\pi^{N_l}, P^{\epsilon'}}(\hat{\pi}), \theta^* - \hat{\theta}\rangle \\
&\leq \|\phi^{\pi^{N_l}, P^{\epsilon'}}(\pi^*) - \phi^{\pi^{N_l}, P^{\epsilon'}}(\hat{\pi})\|_{\hat{\Sigma}_{N_l}^{-1}} \|\theta^* - \hat{\theta}\|_{\hat{\Sigma}_{N_l}} \qquad (\hat{\Sigma}_{N_h}^{-1} \preceq \hat{\Sigma}_i^{-1}) \\
&\leq \frac{1}{N_h} \sum_{i=1}^{N_h} \|\phi^{P^{\epsilon'}}(\pi_1^i) - \phi^{P^{\epsilon'}}(\pi_2^i)\|_{\hat{\Sigma}_i^{-1}} \|\theta^* - \hat{\theta}\|_{\hat{\Sigma}_{N_l}} \qquad (\diamond) \\
&\leq \frac{1}{\sqrt{N_l}} (2d\log(1 + \frac{N_l}{d})) \|\theta^* - \hat{\theta}\|_{\hat{\Sigma}_{N_l}}
\end{aligned}
$$

$(\diamond)$ : since low-level policy feature expectation is a superset of high-level policy expectation, it follows that by choice of low-level policies $\pi_1^i, \pi_2^i$: $\|\phi^{P^{\epsilon'}}(\pi_1^i) - \phi^{P^{\epsilon'}}(\pi_2^i)\|_{\hat{\Sigma}_i^{-1}} \geq \|\phi^{\pi^{N_l}, P^{\epsilon'}}(\pi^*) - \phi^{\pi^{N_l}, P^{\epsilon'}}(\hat{\pi})\|_{\hat{\Sigma}_{N_l}^{-1}}$

Moreover, since low-level feedback is always unbiased, $\|\theta^* - \hat{\theta}\|_{\hat{\Sigma}_{N_l}} = O(1)$. Thus, the overall rate comes out to $O(N_l^{-1/2})$.

**Remark 4** (High Probability Guarantee). *For completeness, we show that the theorem statement holds with probability at least $1 - \delta$:*

$$
\begin{aligned}
&\Pr(\bigcap_{s,a} \mathcal{E}^{s,a} \cap \mathcal{E}_1^l \cap \mathcal{E}_2^l \cap \mathcal{E}_1^h \cap \mathcal{E}_2^h) \\
&\geq 1 - \Pr(\neg \bigcap_{s,a} \mathcal{E}^{s,a}) - \Pr(\neg \mathcal{E}_1^l) - \Pr(\neg \mathcal{E}_2^l) - \Pr(\neg \mathcal{E}_1^h) - \Pr(\neg \mathcal{E}_2^h) \\
&\geq 1 - \delta/5 - \delta/5 - \delta/5 - \delta/5 - \delta/5 \\
&= 1 - \delta
\end{aligned}
$$

### D.2.7 Additional Guarantees

In addition, we derive requisite conditions on the constants for idealized-feedback (the most interesting case).

**Necessary Auxiliary Parameters Bound:** We have that,

$$H_h(\frac{C_1}{\sqrt{N_l}} + C_2\epsilon') + 2C(\epsilon',\delta) + \frac{1}{\sqrt{N_h}}(2d\log(1 + \frac{N_h}{d}))\|\theta^* - \hat{\theta}\|_{\hat{\Sigma}_{N_h}}$$

$$\leq H_h(\frac{C_1}{\sqrt{N_l}} + C_2\epsilon') + 2C(\epsilon',\delta) + N_h^{-1/2}2d\left(2\|\theta^* - \theta^{N_h}\|_{\hat{\Sigma}_{N_h}} + O(B\sqrt{d\log N_h/\delta}W) + \sqrt{8N_h}C(\epsilon',\delta)\right)$$

$$\leq H_h(\frac{C_1}{\sqrt{N_l}} + C_2\epsilon') + (8d+2)C(\epsilon',\delta) + N_h^{-1/2}2d\left(\left(\frac{N_h^{1/2}}{N_l^{1/4}} + (N_h\epsilon')^{1/2}\right) + C' + O(B\sqrt{d\log N_h/\delta}W)\right)$$

$$\leq (H_hC_1)N_l^{-1/2} + 2dN_l^{-1/4} + C_2H_h\epsilon' + d\epsilon'^{1/2} + 9dC(\epsilon',\delta) + 2dC''N_h^{-1/2}$$

$$= (H_hC_1)N_l^{-1/2} + 2dN_l^{-1/4} + C_2H_h\epsilon' + d\epsilon'^{1/2} + 9d\left(2H_hBRd^2\epsilon' + 8H_h^3H_l\delta\right) + 2dC''N_h^{-1/2}$$

$$\leq (2d + H_hC_1)N_l^{-1/4} + (C_2H_h + 18d^3H_hBR)\epsilon' + 72dH_h^3H_l\delta + 2dC''N_h^{-1/2}$$

Setting the upper bound to be below $\epsilon$, or each term to be below $\epsilon/4$, we obtain the following bounds:

- $N_l \geq O(\frac{(d + H_hC_1)^4}{\epsilon^4})$.

- $N_h \geq O(\frac{d^2}{\epsilon^2})$.

- $\kappa \geq O(\frac{dH_h^2H_l}{\epsilon})$:
  $72dH_h^3H_l\delta \leq \epsilon/4 \Rightarrow \delta \leq O(\frac{\epsilon}{dH_h^3H_l})$.
  Recall $\delta = \frac{1}{\kappa H_h} + \frac{C_1}{\kappa H_hH_l\sqrt{N_l}} + \frac{C_2\epsilon'}{\kappa H_hH_l} + H_l\epsilon'$.
  This implies that $\kappa \geq O(\frac{dH_h^2H_l}{\epsilon})$ and $\epsilon \leq O(\frac{\epsilon}{dH_h^3H_l^2})$.

- $\epsilon' \leq O(\min(\frac{\epsilon}{dH_h^3H_l^2}, \frac{\epsilon}{d^3H_hBR}))$:
  Finally, we also require that $(C_2H_h + 18d^3H_hBR)\epsilon' \leq \epsilon/4 \Rightarrow \epsilon' \leq O(\frac{\epsilon}{d^3H_hBR})$. Thus, we need that $\epsilon' \leq O(\min(\frac{\epsilon}{dH_h^3H_l^2}, \frac{\epsilon}{d^3H_hBR}))$.

# E    Statistical Efficiency of HRL

An useful sanity check for hierarchical RL algorithms is that it achieves improved statisical sample complexity in settings with repeated sub-MDP structure [25]. As in [25], we examine if Algorithm 1 also improves upon algorithms that do not leverage hierarchical structure. We make this comparison with vanilla UCB-VI under the same isomophism assumption.

**Corollary 5.** *Setting $\mathcal{A}_{s,a}$ to be the standard UCB-VI algorithm with $UB(\mathcal{R}^{N^{K,H_h}(s,a)}) = O(H_l^{3/2}\sqrt{|S_{s,a}^l||A|N^{H_h,K}(s,a)})$, we have the following bound:*

$$\sum_{s,a \in \mathcal{C}(S,A)} UB(\mathcal{R}^{N^{K,H_h}(s,a)}) + H^h H^l \sqrt{N^{K,H_h}(s,a)}$$

$$\leq \tilde{O}(H_l^{3/2}\sqrt{\max_{s,a}|S_{s,a}^l||A|}\sqrt{|\mathcal{C}(S,A^h)|(H_h K)} + H_h H_l \sqrt{|\mathcal{C}(S,A^h)|H_h K})$$

**Comparison with vanilla UCB-VI:** Standard application of UCB-VI yields the following rate: $\tilde{O}((H_h H_l)^{3/2}\sqrt{|S||A|K})$. Hier-UCB-VI compares favorably to vanilla UCB-VI, if $\max_{s,a} |S_{s,a}^l||\mathcal{C}(S,A^h)| << |S|$. Or in words, there is a lot of repeated/identical sub-MDPs and sub-MDPs have small state space size.

