# OpenReview forum: "A theoretical case-study of Scalable Oversight in Hierarchical Reinforcement Learning"
_NeurIPS.cc/2024/Conference — NeurIPS 2024 poster_

### Official Review · Reviewer_z273 · 2024-07-09

**Soundness:** 3
**Presentation:** 2
**Contribution:** 2
**Rating:** 5
**Confidence:** 3

**Summary:**

The paper studies the scalable oversight problem in the goal-conditioned hierarchical reinforcement learning setup. The starting point of the paper is that, if the time horizon $H$ is large, there is only a limited amount of feedback that can be given (authors use an example of essay or code writing, which requires the labeller to read through and provide feedback on the whole output, which is is very expensive). Thus, it is crucial to leverage hierarchical feedback (e.g. judging one paragraph, or one helper function, or the high-level plan). Given a tabular MDP, the task of finding a good policy is decomposed into two levels:
 - a high-level policy $\pi^h$ that computes, in a state $s$, a high-level action $a^h$ which in turn gives a goal $g(s,a^h) \in S$ and a sub-MDP $M(s, a^h)$ that operates on a subset of states $S_{s, a^h} \subseteq S$, but has access to all actions $A$
 - a set of low-level policies $\pi_{s,a^h}$, for each state $s$ and high-level action $a^h$, which attempt to reach a goal $g(s,a^h)$ (almost surely) while collecting as much reward on the way as possible.

In the first part of the paper, the feedback is assumed to be cardinal. The goal function is assumed to be given and fixed. The task studied is then to find a high-level policy and a set of low-level policies, which give close-to-optimal return. This is done by employing a two-level UCB-VI procedure, where high-level actions correspond to learning UCB over sub-MDPs. The reward feedback is then only given for the low-level trajectories of length $H_l$ (since the high-level reward is computed by adding low-level ones).

In the second part of the paper, the feedback is assumed to be ordinal, and to follow BTL model. Authors point out that there is a subtle issue with the low-level feedback not always being sufficient for no-regret learning, and high-level feedback depending on what the labeller believes the low-level polices will be (e.g. whether they judge wrt to optimal or actual low-level policies). They develop a hierarchical preference-learning algorithm H-REGIME and analyse its regret in all three cases discussed above.

**Strengths:**

The paper studies an important problem of scalable oversight, and approaches it from a perspective that could potentially be used as a basis for a practical implementation, both in ordinal and cardinal setup, at least if the problems mentioned in Weaknesses are resolved. The fundamental idea of doing a hierarchical UCB-VI is quite simple, but computing the right form of the bonus, and the insight to operate in the goal-conditioned setup (leading to a particular form of low-level reward functions and simplifying the theory quite a lot) seem novel and non-trivial. The paper looks to be technically sound, and although I did not carefully check very technical proofs of various inequalities, which span 30 pages in the appendix, very detailed notes make it possible to spot-check it without issues.

**Weaknesses:**

My main question/objection is about the H-UCB-VI algorithm. For shared learning, the algorithm involves assigning the chosen sub-MDP to an appropriate cluster $C(s,a)$. This means that the transition probabilities and rewards must be known for the whole MDP (well, for all sub-MDPs). But if this is the case, then there's no need for using UCB - we can use VI directly. Moreover, the paper improves over UCB-VI only in case there are many (small) isomorphic sub-MDPs, which makes the dependence on shared learning crucial for this to be useful. (Moreover, the motivating example used throughout the paper is given to be essay writing, which does not seem to enjoy this property. Other than that, authors do not give any other examples, nor do any concrete example calculations - even in the appendix - which makes it more difficult to understand and appreciate the work.) This is where I might be fundamentally misunderstanding the work - please correct me if I'm wrong, this is the main reason for my low rating.

In general, it was very unclear to me what exactly is the influence of the goal function on the whole setup - see questions below. The discussion was limited to a few lines in "Goal Selection" paragraph, but I did not gain a good intuition the interplay between the high-level policy performance, and the goal function. (For example, how much work is good goal function doing when comparing to the baseline UCB-VI.)

**Questions:**

Questions:
 - The definition of the MDP in the introduction does not contain a reference to high-level actions. How do they fit into the definition, formally?
 - Why are states for sub-MDPs indexed by high-level action as well? Why are all actions $A$ available in sub-MDPs? What if an action leads outside of $S_{s,a}$?
 - In what way do results depend on the goal function? Can it be quantified? Is it possible to derive $W(g)$ of the value of the optimal policy wrt to this goal function, and say something about the sensitivity of of this $W$ on the parameter? How would goal functions be derived in a realistic setup? (Those questions are probably somewhat, or even significantly, out of scope of the paper, but I'd appreciate if the authors provided at least some preliminary answers, since the practicality and a lot of the value of this development hinges on this problem).
 - What are the policy classes $\Pi^h$ and $\Pi_{s,a}$? They appear to be assumed in the definitions, but are not introduced otherwise.
 - I do not know the REGIME algorithm, but I could not find it in the provided reference?
 - Does the approach discussed in the paper extend to more than two levels of hierarchy?
 - What is $S_h$ in line 217/218?
 - Algorithm 2, line 12 - shouldn't it say length H^h trajectories?

**Limitations:**

Yes.

---

> ### Author Rebuttal · Authors · 2024-08-07
>
> Thank you very much for your detailed review, reviewer z273! These are thoughtful questions and we address the main concerns below due to the space limit. We will be sure to clarify the notation (e.g. $\Pi^h$) as you suggest.
>
> ```
> …The task studied is then to find a high-level policy and a set of low-level policies…This is done by employing a two-level UCB-VI procedure…
> ```
>
> Thanks for the thorough summary! Just to clarify, H-UCB-VI does not have to use UCB-VI as a subroutine. Any no-regret subroutine will do (L177). For instance, if sub-MDPs are linear, one can use LSVI. If sub-MDPs are linear and share structure, one can use more specialized algorithms like [11] for better guarantees (L220-2).
>
> ```
> My main question/objection is about the H-UCB-VI algorithm. For shared learning, the algorithm involves assigning the chosen sub-MDP to an appropriate cluster. This means that the transition probabilities and rewards must be known for the whole MDP… But if this is the case, then there's no need for using UCB - we can use VI directly. Moreover, the paper improves over UCB-VI only in case there are many (small) isomorphic sub-MDPs, which makes the dependence on shared learning crucial for this to be useful.
> ```
>
> Thanks for this question:
>
> 1. Firstly, the assumption from [25] is that we know which sub-MDPs share the same transitions and belong to the same cluster. It does not mean we know the transitions of the MDPs in the clusters, as this would indeed trivialize the problem (and the results of [25]).
> 2. Next, the concern is that the paper improves upon UCB-VI when there are many isomorphic sub-MDPs. Our goal in providing this corollary (of our main result) is simply to emulate the analysis of [25], which makes this assumption to show the statistical efficiency of HRL. We do not claim that this assumption is widely applicable. Rather, this corollary serves a “sanity check” that our hierarchical algorithm *is* more efficient in a setting where it should be (L211-213).
> 3. Our main result in this section is that HRL reduces to multi-task, sub-MDP regret minimization. This result allows one to flexibly leverage shared structure, beyond the cluster assumption, to improve the regret. Please see our answer above.
>
> If it improves the presentation of the paper, we can move the corollary to the appendix and not mention the clusters to avoid confusion, as this corollary is not central to the paper.
>
> ```
> …authors do not give any other examples, nor do any concrete example calculations - even in the appendix - which makes it more difficult to understand and appreciate the work.)
> ```
>
> Yes, we can definitely provide more examples to aid understanding! A canonical example in HRL is the maze (e.g. [22]). A maze consists of rooms with doors. The goal is to get to the exit in as few steps as possible.
>
> 1. For the global MDP, $S = S^h \times S^l$ where $s^h$ denotes the index of the current room, and $s^l$ denotes the position of the agent in the room. Action set $A$ consists of moving (L, R, U, D, Stay).
> 2. High-level MDP: high-level action $A^h$ consists of moving to the (N, S, E, W) door of the room. $s$ is the current location of the agent, and $g(s, a^h)$ maps the goal (door) to its location.
> 3. Low-level MDP: has state space $S^l_{s,a} \subset S$ and the action set $A$ is the same moving (U, D, L, R, Stay).
>
>
> ```
> The definition of the MDP in the introduction does not contain a reference to high-level actions. How do they fit into the definition, formally?
> ```
>
> The high-level action $A_h$ is usually set based on prior knowledge. We describe what it is in the essay example (L63-5). $A_h$ does not have to be related to $A$ in the global MDP definition. For example in the maze case, moving to the N,S,E,W door is a wholly different action set from moving in the U,D,L,R direction.
>
> ```
> Why are states for sub-MDPs indexed by high-level action as well? Why are all actions $A$ available in sub-MDPs? What if an action leads outside of $S_{s,a}$?
> ```
>
> The high level action is needed to determine the goal state of the sub-MDP, which defines the sub-MDP reward. The action set is defined to be $A$ in sub-MDPs, as is commonly assumed in GC-HRL literature. An action cannot lead to a state outside of $S^l_{s,a}$, because by definition, $S^l_{s,a}$ is the set of all reachable states (L60).
>
> ```
> In what way do results depend on the goal function? Can it be quantified?…How would goal functions be derived in a realistic setup?
> ```
>
> Thanks for this question, we agree that our learned policy is only as good as the goal function chosen (L224-5), which is key to the success of GC-HRL:
>
> 1. As we write on L29-32, there are already many settings of interest where we *have* prior knowledge of a good hierarchy/goal function. This is because we humans have often (and successfully) taken the hierarchical approach to build up to and produce these long-form creations. So we know what are good goals to set e.g. we write essays by first writing an outline of arguments, then expanding out each point in the outline.
> In such settings, the algorithms we develop can already help to scale up bounded feedback and enable scalable oversight.
> 2. Indeed, this approach of explicitly encoding prior knowledge in the learning algorithm is common in both GC-HRL literature (e.g. we know apriori mazes consist of rooms [22]) and scalable oversight literature (e.g. books consist of chapters [27]).
> 3. Outside of such settings, we agree it is an open problem to learn apt hierarchical decompositions/goals (L329-30). It is an exciting direction that could realize end-to-end scalable oversight [15].
>
> ```
> Does the approach discussed in the paper extend to more than two levels of hierarchy?
> ```
>
> Our algorithm is applicable to any number of levels of hierarchy due to the reduction of HRL to multi-task, sub-MDP regret minimization. This allows for the case when each sub-MDP is hierarchical as well, and one can invoke H-UCB-VI calling H-UCB-VI as the subroutine.

---

> > ### Comment · Reviewer_z273 · 2024-08-12
> > **Response**
> >
> > I thank the authors for their thorough response. I think all of the minor points and questions are satisfactorily handled, except I still don't understand how exactly the authors want the set $A_h$ to be formally defined (e.g. [25] introduces a similar notion formally in Definition 1 using a notion of partition of states MDP states).
> >
> > However, I am still unconvinced by the assumption that clustering function is available. I admit it isn't literally implying that transition probabilities and rewards are known, but at the same time, I have hard time imagining any semi-realistic situation in which one does not imply the other. In full RL, the agent doesn't know which situation (which sub-MDP) it found itself in, which would necessitate querying the clustering function online, and that in turn would imply knowing $\tau$ and $R$. Indeed, the running example of essay writing is still not showing that, and the maze environment is very simplistic.
> >
> > As far as I see, [25] does not discuss the feasibility of the assumption about the partition being known (agent having access to the clustering function, in the terminology here), but a large amount of their analysis is in the context of planning, where the full structure of MDP is known anyway.
> >
> > Of course, it is possible that the algorithm would still work as expected if the clustering was only approximate - but the paper does not say anything about that.
> >
> > I am therefore maintaining my rating for now.

---

> > > ### Author Response · Authors · 2024-08-12
> > > **Reply to Reviewer z273**
> > >
> > > Thank you for getting back to us!
> > >
> > > ```
> > > I am still unconvinced by the assumption that clustering function is available. I admit it isn't literally implying that transition probabilities and rewards are known, but at the same time, I have hard time imagining any semi-realistic situation in which one does not imply the other.
> > > ```
> > >
> > > Firstly, we wish to emphasize that this assumption of the known clustering function is *only* used in this paper to derive Corollary 1, to do the "sanity check" comparison. This assumption is *not* central to the paper's main results. Outside of Corollary 1, removing this assumption does not affect any other result. This is because overall, our work is about scaling up bounded feedback via HRL, and not comparing the statistical efficiency of HRL vs RL (as in [25]). We will be sure to clarify this in our revision and we apologize for the confusion.
> > >
> > > Secondly, as we write in our previous response, we agree with you that the cluster assumption from [25] is quite specific/strong. When we remove the assumption and the cluster function is unknown, we can simply have each sub-MDP be its own cluster. In such settings, our main result in section 3 (Theorem 1) is still useful as it can capture improvements in regret from other general forms of shared learning, since we have shown learning reduces to multi-task, sub-MDP regret minimization.
> > >
> > > One example of this is [11] applicable to the linear sub-MDP setting (L221-222), which makes use of the low-rank and not the cluster assumption. This is perhaps another point of difference between our paper and [25] in that we formalize the reduction to multi-task learning, which demonstrates improvements in the regret bound from other forms of shared learning.
> > >
> > >
> > > ```
> > > In full RL, the agent doesn't know which situation (which sub-MDP) it found itself in
> > > ```
> > >
> > > As a point of clarification, in our setting, the agent *does* know which sub-MDP it is in. sub-MDPs are defined by the state and high-level action, both of which are known to the agent. Presumably, you meant to write "sub-MDP cluster" here?
> > >
> > > ```
> > > ...except I still don't understand how exactly the authors want the set $A_h$ to be formally defined (e.g. [25] introduces a similar notion formally in Definition 1 using a notion of partition of states MDP states).
> > > ```
> > >
> > > Thank you for this! The definition you point to in [25] is specific to a particular class of hierarchical MDPs. We chose not to define $A_h$, because it is really dependent on the MDP's hierarchical structure. But we agree with you that for concreteness it helps to have a formal definition. Following your suggestion, we will cite their definition as an example of how $A_h$ can be formally defined to model a certain class of hierarchical MDPs (e.g. mazes).
> > >
> > > Thank you again for taking the time, and please do not hesitate to follow up if you have any further questions! We appreciate it.

---

> > > > ### Comment · Reviewer_z273 · 2024-08-12
> > > > **Response**
> > > >
> > > > I thank the authors for the clarification.
> > > >
> > > > I'd like to continue this discussion for a bit still. In light of the above, it seems that I have trouble interpreting the content of the Theorem 1. My understanding is that:
> > > >  - Theorem 1 analyses the behavior of Algorithm 1, and shows that the regret grows not faster than (number of clusters)*(upper bound on regret in a cluster).
> > > >  - As the authors claim above, Algorithm 1 can work with a clustering function that is a refinement of the true clustering function (that is, the clusters are more fine-grained).
> > > >  - However, as far as I understand, the bound uses the $C$ that is actually used in the algorithm, and not the (unknown) "ground-truth" clustering given by Definition 2.
> > > >  - In particular, if we assume no prior knowledge of the clustering structure, the bound becomes proportional to $|A_h|\cdot |S|$
> > > >  - I don't know what is the cardinality of the set $A_h$ is - in particular, I am not sure what is the "MDP's hierarchical structure" the authors refer to in the above answer. But my reading was that, in this case, it was trivialising the bound of Theorem 1. (I.e. if the hierarchical structure is a prior unknown, then we can do no better than a normal non-hierarchical approach).
> > > >
> > > > (Also, on a related note, it is probably advisable to write explicitly the chain of regret inequalities
> > > >
> > > > $\sum_{k=1}^{K} V^{\pi^*}_1(s_1) - V^{\pi_k}_1(s_1) \leq \sum V^{k}_1(s_1) - V^{\pi_k}_1(s_1) \leq \ldots$
> > > >
> > > > which follows, as far as I understand, from the optimism lemma in the Appendix B.2. Otherwise, it is not clear what the exact connection to regret even is, since $V^{k}_1$ is not analysed before this point. And there appears to be some typography problem in the statement of the optimism lemma.)

---

> ### Author Response · Authors · 2024-08-12
> **Reply to Reviewer z273**
>
> Thanks for continuing to engage with us! We are happy to go over this, and will be sure to correct the typo as you suggest, thanks.
>
> ```
> Theorem 1 analyses the behavior of Algorithm 1, and shows that the regret grows not faster than (number of clusters)*(upper bound on regret in a cluster).
> ```
>
> More precisely, it is the sum of the upper bound on the regret of the clusters, which in settings with shared structure between sub-MDPs can be better than (number of clusters)*(upper bound on regret in a cluster).
>
> ```
> As the authors claim above, Algorithm 1 can work with a clustering function that is a refinement of the true clustering function (that is, the clusters are more fine-grained).
> ```
>
> That is right, Algorithm 1 provides a regret guarantee for any cluster function that is correct.
>
> Our previous point is that even in settings where we take each sub-MDP to be its own cluster (the most refined clustering if you'd like), improvements in the regret bound are still possible e.g. due to low-rank structure.
>
> ```
> However, as far as I understand, the bound uses the $C$ that is actually used in the algorithm, and not the (unknown) "ground-truth" clustering given by Definition 2.
> ```
>
> This is correct. When the cluster function is unknown, the algorithm will use the one where each sub-MDP is its own cluster (and not the unknown, best cluster function).
>
> ```
> In particular, if we assume no prior knowledge of the clustering structure, the bound becomes proportional to $|A_h|\cdot |S|$... my reading was that, in this case, it was trivialising the bound of Theorem 1. (I.e. if the hierarchical structure is a prior unknown, then we can do no better than a normal non-hierarchical approach).
> ```
>
> That’s right, without prior knowledge, we do not do better than the non-hierarchical algorithm in terms of statistical complexity, as you write. With that said, Corollary 1 emulates the comparison done in [25]. In [25], it is assumed this cluster function is *known* and the HRL algorithm is proven to be better under certain conditions. We do the same here, also assuming the cluster function is known (with no endorsement of how realistic this assumption is). We agree with you it would be interesting future work to design an algorithm that can cluster on the fly.
>
> Finally, and importantly, the comparison above is in terms of statistical efficiency. Under our assumption of bounded feedback, the non-hierarchical approach is actually *not applicable* here, due to its excess trajectory length of $H_h H_l$ (L54). Hence, we believe Algorithm 1 is not trivial, but rather a useful contribution that can achieve no-regret while learning from bounded cardinal feedback.
>
> Again, thank you for your time and please let us know if there are any other questions!

---

> > ### Comment · Reviewer_z273 · 2024-08-13
> > **Response**
> >
> > I again applaud the authors for engaging with me on this.
> >
> > I think the last exchange validates the key points of my understanding of the theorem. So I remain convinced about the centrality of the known-clustering assumption, via the chain "Unknown clustering" -> "Algorithm 1 uses trivial clustering" -> "Theorem 1 proves trivial bound about its regret" -> "Corollary 1 doesn't improve on the baseline".  Or, more straightforwardly, if the learner doesn't know that sub-MDP A is isomorphic to a sub-MDP B, but they indeed are, presented results are not impactful in any way.
> >
> > At the same time, I agree that, introducing the assumption about the feedback being limited to $H_l, H_h$ is an important (but, I think, a bit separate, point), which makes the contribution valuable on its own, so I decided to increase my score to recommend acceptance.

---

> > > ### Author Response · Authors · 2024-08-13
> > > **Reply to Reviewer z273**
> > >
> > > Thank you for your timely response and update! To wrap up our discussion, we have two more quick points to share if we may.
> > >
> > > ```
> > > So I remain convinced about the centrality of the known-clustering assumption, via the chain "Unknown clustering" -> "Algorithm 1 uses trivial clustering" -> "Theorem 1 proves trivial bound about its regret" -> "Corollary 1 doesn't improve on the baseline".
> > > ```
> > >
> > > This is exactly right. To confirm, without this assumption, we cannot derive Corollary 1. The clustering assumption is central to the favorable comparison result (and that of [25]).
> > >
> > > ```
> > > if the learner doesn't know that sub-MDP A is isomorphic to a sub-MDP B, but they indeed are, presented results are not impactful in any way.
> > >
> > > At the same time, I agree that, introducing the assumption about the feedback being limited to $H_l, H_h$ is an important (but, I think, a bit separate, point), which makes the contribution valuable on its own
> > > ```
> > >
> > > Just to offer our understanding,
> > >
> > > 1. Under the “trivial clustering”, Theorem 1 leads to some no-regret guarantee. In a vacuum, we completely agree with you that this guarantee is not impactful. It doesn’t improve upon the baseline (non-hierarchical algorithm), which one can readily use.
> > > 2. However our previous point was that under the premise of the paper (limited feedback), the regret guarantee, even under the “trivial clustering”, does become *more* meaningful. This is because the baseline is no longer applicable. And so, we think having an algorithm that can achieve *some* no-regret guarantee is a step forward, in the context of scalable oversight.
> > >
> > > Lastly, we definitely agree that the algorithm can be improved, since it doesn’t do online clustering. It can only leverage certain types of multi-task structure, but not all. We will highlight this as a key direction for future work.
> > >
> > > Please let us know if there are more questions, and thank you again for your thorough engagement during the process! It has definitely helped to improve our paper.

---

### Official Review · Reviewer_poPw · 2024-07-11

**Soundness:** 3
**Presentation:** 3
**Contribution:** 2
**Rating:** 6
**Confidence:** 2

**Summary:**

Provides proofs of the regret bounds for cardinal and ordinal feedback using the sub-MDP framework of goal-conditioned HRL. Sub-MDPs are defined by a starting state, fixed horizon, high-level action and subspace. The high-level policy selects transitions between sub-MDPs. This work first proposes an upper confidence bound algorithm based on lower-level reachability and selection, then proves the regret of this algorithm.

**Strengths:**

The method provides clear and applicable analysis of hierarchical goal-conditioned RL, even outside of the context of feedback.

The derivation of the regret due to goal-conditioned HRL is informative and interesting.

The proposed insight of ensuring hierarchical consistency is meaningful for many applications of GCHRL.

**Weaknesses:**

Algorithm 1 is somewhat difficult to follow. In particular, while the components make sense, the core insight of the bounds on the lower-level goals is difficult to locate.

The regret analysis proof sketch could provide more insight. In particular, the separation of which components occur because of the high-level and low-level errors, which introduce the product of horizons, is not intuitively clear.

The work for ordinal feedback is not particularly well contained, since it relies on the properties of REGIME, which are not made clear in this work. In particular, it is not obvious without looking into that work where the sub-policy simulator fits in.

In many ways this seems less to be a work about the nature of scalable feedback, and simply on the nature of hierarchical RL in providing regret bounds. This subject is of interest regardless but calls to question the framing of this work.

**Questions:**

Can the work on regret analysis of GCHRL be separated from the analysis of ordinal rewards?

What is the relationship between the analysis of these two?

**Limitations:**

The analysis of other components of HRL, such as goal sampling, learning procedure, etc. are not that well captured, though this is typically challenging to analyze theoretically.

---

> ### Author Rebuttal · Authors · 2024-08-07
>
> Thank you very much for your detailed review, reviewer poPw! These are great questions, which have definitely helped to improve our paper and its presentation.
>
> ```
> Algorithm 1 is somewhat difficult to follow. In particular, while the components make sense, the core insight of the bounds on the lower-level goals is difficult to locate.
>
> The regret analysis proof sketch could provide more insight. In particular, the separation of which components occur because of the high-level and low-level errors, which introduce the product of horizons, is not intuitively clear.
> ```
>
> Thank you for this question! To answer your question, the regret is *only* a function of low-level error in the cardinal case. Hence, one of our main results is that GC-HRL reduces to multi-task, sub-MDP regret minimization. In our proof sketch, we provided an outline of the errors terms that contribute to regret. More specifically:
>
> 1. $\rho_h^k$: regret due to low level policy incurring sub-optimal returns.
> 2. $\gamma_h^k$ and $\sigma_h^k$: regret due to sub-optimal low-level policy not reaching the goal state.
>
> The product of horizons $H_h H_l$ shows up in the low-level regret when low-level policies miss the goal. Indeed, when the goal state is not reached, it is unclear which state the agent would be in. Thus, the maximal difference between any two returns, $H_h H_l$, is used as an upper bound on the resultant regret.
>
>
> ```
> The work for ordinal feedback is not particularly well contained, since it relies on the properties of REGIME, which are not made clear in this work. In particular, it is not obvious without looking into that work where the sub-policy simulator fits in.
> ```
>
> Thank you for this nice feedback! Our current algorithm includes the full details of REGIME. We agree with you that a more modular description of Algorithm 2 would improve the presentation. We will be sure to include this in our next revision. Here it is with reference to line numbers:
>
> > 1. Invoke one copy of REGIME *across* all sub-MDPs with shared exploration (L2-4) and learned reward (L6).
> > 2. Compute near-optimal, sub-MDP policies $\pi^{N_l}_{s,a}$ that the high level policy will invoke (L7).
> > 3. Invoke one copy of REGIME for the high-level MDP, where the feature expectation of each sub-MDP is computed according to $\pi^{N_l}_{s,a}$ and $P^{\epsilon’}$.
>
> We arrived at Algorithm 2 by improving upon the naive application of REGIME to the hierarchical setting, with the improvements as describe on L295-307.
>
> Finally, as for the role of the simulator, the REGIME algorithm assumes access to a simulator $P^{\epsilon’}$ to compute policy feature expectations. We do the same when invoking REGIME, denoting the feature expectation with notation $\phi^{P^{\epsilon’}}(\pi)$ in our algorithm.
>
>
> ```
> In many ways this seems less to be a work about the nature of scalable feedback, and simply on the nature of hierarchical RL in providing regret bounds. This subject is of interest regardless but calls to question the framing of this work.
> ```
>
> To clarify, our work does shed some light on the nature of scalable feedback. We show that one way to generate scalable feedback is by leveraging hierarchical structure (L27-29). Our work analyzes how to scale feedback, and develops HRL algorithms that efficiently learn with provable guarantees.
>
> A general point that our paper makes is that besides the benefits of easier credit assignment and exploration (as listed in [25]), HRL has the added benefit of allowing for scalable oversight.
>
> Certainly, we agree that our work does not provide a full characterization of “the nature of scalable feedback”, as you write. We study one natural setting (hence a “case-study” as in our title), the hierarchical setting, where we show we can provably scale up bounded feedback.
>
> This may not be the only setting that allows for scalable feedback. And discovering other types of scalable feedback and analyzing their nature is verily unexplored (and exciting) territory.
>
>
>
> ```
> Can the work on regret analysis of GCHRL be separated from the analysis of ordinal rewards?
>
> What is the relationship between the analysis of these two?
> ```
>
> Thanks for this question! The key difference between the two analyses is that in the cardinal case, the total regret decomposes into the sum of sub-MDP regret (Theorem 1).  In the ordinal case, the total regret decomposes into both high-level MDP regret and sub-MDP regret. As we show in Proposition 1, this is because we need to not only learn from comparison which policies are best within sub-MDPs (incurring low-level regret), but also which sub-MDPs yield the highest returns across sub-MDPs (incurring high-level regret). This is made explicit starting from the third line of the proof of Theorem 4 (L645 in the appendix), and results in a different type of analysis.
>
> The key commonality underlying the two analyses is that an apt sub-MDP reward design is needed to incentivize goal-reaching *in balance* with maximizing the return of the sub-MDP (L172-175). This we believe is one of our main contributions, in finding the “right” reward setting and weighting to balance the two objectives. It is also what allows us to derive a new form of bonus that trades off between the two (L176-8), such that we can bound the resultant cumulative regret w.r.t. the global MDP.

---

> > ### Comment · Reviewer_poPw · 2024-08-09
> > **Response to authors**
> >
> > I appreciate the detailed response and believe the work should be accepted, and will maintain my score.

---

> > > ### Author Response · Authors · 2024-08-09
> > > **Reply to Reviewer poPw**
> > >
> > > Thank you for getting back to us, and we appreciate your helpful feedback!

---

### Official Review · Reviewer_Hzkp · 2024-07-13

**Soundness:** 3
**Presentation:** 2
**Contribution:** 3
**Rating:** 6
**Confidence:** 2

**Summary:**

This paper analyzes scalable oversight in the context of goal-conditioned hierarchical reinforcement learning. Specifically, it theoretically shows that it is possible to efficiently use hierarchical structure to learn from bounded human feedback.

**Strengths:**

* The problem is significant and of high importance to the community.
* The theoretical analysis and sub-MDP reward design for Hierarchical-UCB-VI appears novel. Furthermore, the extension to ordinal (preferences) feedback appears novel, specifically in the proposed Hierarchical-REGIME algorithm and analysis.

**Weaknesses:**

* Minor: It would be a nice addition to the paper to empirically demonstrate Algorithm 1 &2 as well.
* Minor: Clarity: Paper had quite a few typo’s. I would encourage the authors to use an automated tool to check for spelling / grammar mistakes.

Typos:

* L61: “policies aims to” -> “policies aim to”
* L183: “approach is avoid” -> “approach is to avoid”
* L218: “identifical” -> “identical”
* L250: “what can we assume the” -> “what can we assume is the”

**Questions:**

* Does the analysis for the linear setting generalize to the non-linear / function approximation setting?

**Limitations:**

Yes they are discussed in Section 5.

---

> ### Author Rebuttal · Authors · 2024-08-07
>
> Thank you for your review, reviewer Hzkp! It has definitely helped to improve our paper and its presentation.
>
> ```
> Paper had quite a few typo’s. I would encourage the authors to use an automated tool to check for spelling / grammar mistakes.
> ```
>
> Thank you for your careful reading! We will be sure to correct these.
>
> ```
> Does the analysis for the linear setting generalize to the non-linear / function approximation setting?
> ```
>
> Thanks for this question! Our paper invokes REGIME as a subroutine, taking its guarantees as a given. REGIME is designed with the assumption of linear reward, and we believe more work is required to extend it to the non-linear setting.
>
> With that said, linearity is an often used assumption in offline RLHF literature [29,30]. The justification for it is that we often do not have to learn features $\phi$. Such features may already be available from (self-supervised) pre-training. Thus, for reward modeling, it is common to assume a linear model on top of non-linear features $\phi$ to do reward modeling.
>
> Also, we note that the linearity assumption pertains to the specifics of the subroutine invoked (in this case REGIME). The insight from our analysis of favoring low-level learning given sufficient coverage applies regardless.

---

> > ### Comment · Reviewer_Hzkp · 2024-08-12
> >
> > Thank you for your detailed rebuttal response. I have raised my score.

---

> > > ### Author Response · Authors · 2024-08-12
> > > **Reply to Reviewer Hzkp**
> > >
> > > Thank you for getting back to us, we appreciate it!

---

### Official Review · Reviewer_2zMJ · 2024-07-20

**Soundness:** 3
**Presentation:** 3
**Contribution:** 3
**Rating:** 6
**Confidence:** 3

**Summary:**

In this paper, the authors study how to scale human feedback, in the context of goal-conditioned hierarchical reinforcement learning. For this work, the authors assume that humans can only provide feedback for outputs with length below a certain threshold. Thus, it is necessary to scale the little feedback provided, in order to improve the complete output. This means, for example, that a human can provide feedback about the high-level actions taken in the hierarchy, but not the detailed steps of the low-level policies. The authors propose two algorithms: one that learns from low-level feedback only, and one that learns by incorporating human preference over high-level trajectories. The authors also present regret bounds for the proposed algorithms.

**Strengths:**

The paper studies a combination of very important topics: theoretical understanding of hierarchical reinforcement learning and the outcomes of incorporating human feedback. I believe that the field still lacks a strong theoretical understanding of HRL, and this paper provides a step in that direction.

The paper is generally very well written and easy to follow. The authors clearly state the setting they are dealing with, provide a useful running example, and intuitive discussions about their theoretical results.

**Weaknesses:**

The results are based on a very strong assumption that there is a well-defined goal function available (that is, a goal function that only proposes feasible goals). It is very hard to guarantee goal feasibility, so assuming that this function is given is very ambitious. However, the authors do recognize the limitation of this assumption and propose, on a high-level, how this could be dealt with in future work.

When discussing the bounds related to the second algorithm (Section 4), it was necessary to introduce several constants, but they were not discussed in more detail, providing, for example, intuitions on the magnitude of such constants. Without it, it is more challenging to understand the significance of the bounds.

**Questions:**

There were a few variables/subscripts used throughout the paper that, I believe, were not formally introduced. As some examples, $a$ on line 60, $h$ and $i$ on the Interaction Protocol paragraph (no line numbers here), $l$ on line 67, $\bar{r}$ on line 179, $N(s,a)$ on line 194, $d$ on Lemma 4. Could the authors please add clarifications about these variables?

**Limitations:**

The authors discuss the limitations of the work. No major concerns.

---

> ### Author Rebuttal · Authors · 2024-08-07
>
> Thank you very much for your detailed review, reviewer 2zMJ! It has definitely helped to improve our paper and its presentation.
>
> ```
> The results are based on a very strong assumption that there is a well-defined goal function available (that is, a goal function that only proposes feasible goals). It is very hard to guarantee goal feasibility, so assuming that this function is given is very ambitious.
>
> However, the authors do recognize the limitation of this assumption and propose, on a high-level, how this could be dealt with in future work.
> ```
>
> Thanks for this question, we agree that our learned policy is only as good as the goal function chosen (L224-5), which is key to the success of GC-HRL:
>
> 1. As we write on L29-32, there are already many settings of interest where we *have* prior knowledge of a good hierarchy/goal function. This is because we humans have often (and successfully) taken the hierarchical approach to build up to and produce these long-form creations. So we know what are good goals to set e.g. we write essays by first writing an outline of arguments, then expanding out each point in the outline.
> In such settings, the algorithms we develop can already help to scale up bounded feedback and enable scalable oversight.
> 2. Indeed, this approach of explicitly encoding prior knowledge in the learning algorithm is common in both GC-HRL literature (e.g. we know apriori mazes consist of rooms [22]) and scalable oversight literature (e.g. books consist of chapters [27]).
> 3. Outside of such settings, we agree it is an open problem to learn apt hierarchical decompositions/goals (L329-30). It is an exciting direction that could realize end-to-end scalable oversight [15].
>
>
> ```
> When discussing the bounds related to the second algorithm (Section 4), it was necessary to introduce several constants, but they were not discussed in more detail, providing, for example, intuitions on the magnitude of such constants. Without it, it is more challenging to understand the significance of the bounds…There were a few variables/subscripts used throughout the paper that, I believe, were not formally introduced…Could the authors please add clarifications about these variables?
> ```
>
> Thank you for your careful reading, we will definitely clarify these notations as you suggest! We have also centralized the notation in Table 1 of the appendix, which we will also add to and link in the main paper for added clarity.

---

> > ### Comment · Reviewer_2zMJ · 2024-08-13
> >
> > Thank you for the added details! After reading your response and the other reviews/discussions, I still believe the paper should be accepted.

---

> > > ### Author Response · Authors · 2024-08-13
> > > **Reply to Reviewer 2zMJ**
> > >
> > > Thank you for getting back to us and taking the time to read through everything (which is quite a bit), we appreciate it!

---

### Decision · Program_Chairs · 2024-09-25

**Decision:**

Accept (poster)

**Comment:**

After lengthy discussion, the reviewers are all recommending acceptance of this work. Several reviewers raised valuable questions and concerns regarding the assumptions made, and the clarity of notation and writing---I am recommending the paper for acceptance, but I encourage the authors to take the reviewers' suggestions into account.